EMBO
Molecular Medicine

# Murine *MPDZ*-linked hydrocephalus is caused by hyperpermeability of the choroid plexus

Junning Yang[1], Claire Simonneau[1], Robert Kilker[1], Laura Oakley[2], Matthew D Byrne[2], Zuzana Nichtova[3], Ioana Stefanescu[1], Fnu Pardeep-Kumar[4], Sushil Tripathi[4], Eric Londin[5], Pascale Saugier-Veber[6], Belinda Willard[7], Mathew Thakur[4], Stephen Pickup[8], Hiroshi Ishikawa[9], Horst Schroten[10], Richard Smeyne[2] & Arie Horowitz[1,11,*] (iD)

## Abstract

Though congenital hydrocephalus is heritable, it has been linked only to eight genes, one of which is *MPDZ*. Humans and mice that carry a truncated version of MPDZ incur severe hydrocephalus resulting in acute morbidity and lethality. We show by magnetic resonance imaging that contrast medium penetrates into the brain ventricles of mice carrying a *Mpdz* loss-of-function mutation, whereas none is detected in the ventricles of normal mice, implying that the permeability of the choroid plexus epithelial cell monolayer is abnormally high. Comparative proteomic analysis of the cerebrospinal fluid of normal and hydrocephalic mice revealed up to a 53-fold increase in protein concentration, suggesting that transcytosis through the choroid plexus epithelial cells of *Mpdz* KO mice is substantially higher than in normal mice. These conclusions are supported by ultrastructural evidence, and by immunohisto-chemistry and cytology data. Our results provide a straightforward and concise explanation for the pathophysiology of *Mpdz*-linked hydrocephalus.

**Keywords** cerebrospinal fluid; choroid plexus; hydrocephalus; magnetic resonance imaging; proteomics

**Subject Categories** Genetics, Gene Therapy & Genetic Disease; Neuroscience

## Introduction

Despite strong evidence for the heritability of congenital hydro-cephalus (Munch *et al*, 2012; Kahle *et al*, 2016), to date, only eight genes have been linked to this condition. The earliest monogenic link of hydrocephalus had been made to *L1CAM* (Rosenthal *et al*, 1992; Jouet *et al*, 1993; Van Camp *et al*, 1993; Coucke *et al*, 1994), which encodes the L1 neuronal cell adhesion molecule. Subsequent studies identified *AP1S2*, a gene of subunit 2 of clathrin-associated adaptor protein complex 1 (Saillour *et al*, 2007; Cacciagli *et al*, 2014), and *CCDC88C*, which encodes DAPLE, a protein involved in Wnt signaling (Ekici *et al*, 2010; Drielsma *et al*, 2012; Ruggeri *et al*, 2018). A more recent gene linked to congenital hydrocephalus is *MPDZ*, encoding a large modular scaffold protein that consists of 13 PDZ domains and one L27 domain (Ullmer *et al*, 1998; Adachi *et al*, 2009). Several cases of severe congenital hydrocephalus identified in five consanguineous families (Al-Dosari *et al*, 2013; Saugier-Veber *et al*, 2017) were linked mostly to biallelic nonsense muta-tions that resulted in nonsense-mediated decay and total loss of MPDZ. A milder phenotype in a non-consanguineous family was linked to missense mutations and a heterozygous splice site variant (Al-Jezawi *et al*, 2018). The four genes that have been linked to congenital hydrocephalus most recently, *TRIM71*, *SMARCC1*, *PTCH1*, and *SHH* (Furey *et al*, 2018), regulate ventricular zone neural stem cell differentiation. Their loss-of-function is thought to result in defective neurogenesis, including ventriculomegaly.

Two of the genetically linked congenital hydrocephalus condi-tions in humans were phenocopied in mice. Mice carrying loss-of-function mutations (LOF) in either *L1CAM* (Dahme *et al*, 1997; Rolf *et al*, 2001) or *MPDZ* (Feldner *et al*, 2017) developed severe

1    Cardeza Center for Vascular Biology, Sidney Kimmel Medical College, Thomas Jefferson University, Philadelphia, PA, USA
2    Department of Neuroscience, Sidney Kimmel Medical College, Thomas Jefferson University, Philadelphia, PA, USA
3    Department of Pathology, Anatomy and Cell Biology, Sidney Kimmel Medical College, Thomas Jefferson University, Philadelphia, PA, USA
4    Department of Radiology, Sidney Kimmel Medical College, Thomas Jefferson University, Philadelphia, PA, USA
5    Computational Medicine Center, Sidney Kimmel Medical College, Thomas Jefferson University, Philadelphia, PA, USA
6    Department of Genetics, University of Rouen, Rouen, France
7    Proteomics Core Facility, Lerner Research Institute, Cleveland Clinic Foundation, Cleveland, OH, USA
8    Department of Radiology, University of Pennsylvania Medical School, Philadelphia, PA, USA
9    Laboratory of Clinical Regenerative Medicine, Department of Neurosurgery, Faculty of Medicine, University of Tsukuba, Tsukuba-City, Ibaraki, Japan#
10   Pediatric Infectious Diseases, University Children's Hospital Mannheim, Heidelberg University, Mannheim, Germany
11   Department of Cancer Biology, Sidney Kimmel Medical College, Thomas Jefferson University, Philadelphia, PA, USA
     *Corresponding author. Tel: +1 215 955 8017; E-mail: arie.horowitz@jefferson.edu
     #Correction added online on 10 January 2019 after first online publication: Affiliation 9 was corrected

hydrocephalus similar to human carriers of biallelic *L1CAM* (Kanemura *et al*, 2006) and *MPDZ* mutants (Al-Dosari *et al*, 2013; Saugier-Veber *et al*, 2017). Hydrocephalus results from impediment of the circulation of the cerebrospinal fluid (CSF), causing its accumulation in the brain ventricles as a result of either excessive CSF inflow, attenuated flow through the ventricles, or blocked outflow (Estey, 2016; Kahle *et al*, 2016). The *L1CAM* and *Mpdz* mouse models afforded anatomic and histological analysis for determining the nature of the defects that interfered with CSF circulation. *L1CAM* mice harbor stenosis of the aqueduct of Sylvius between the $3^{rd}$ and $4^{th}$ ventricles, but it was judged to be a result of the increased intraventricular pressure and the ensuing compression of the aqueduct's walls, rather than the cause of hydrocephalus (Rolf *et al*, 2001). The formation of hydrocephalus in $Mpdz^{-/-}$ mouse was attributed to stenosis of the aqueduct (Feldner *et al*, 2017). Postmortem pathology of brains from several individuals harboring LOF *MPDZ* variants detected ependymal lesions but did not reveal causative mechanisms.

MPDZ is a cytoplasmic protein localized close to the junctions of epithelial (Hamazaki *et al*, 2002) and endothelial cells (Ernkvist *et al*, 2009), as well as to neuronal synapses (Krapivinsky *et al*, 2004). In the former cell types, MPDZ binds at least eight junction transmembrane proteins (Hamazaki *et al*, 2002; Poliak *et al*, 2002; Jeansonne *et al*, 2003; Coyne *et al*, 2004; Lanaspa *et al*, 2008; Adachi *et al*, 2009). The abundance of MPDZ in the central nervous system is highest in the choroid plexus (CP) (Sitek *et al*, 2003), a network of capillaries walled by fenestrated endothelial cells, surrounded by a monolayer of cuboidal epithelial cells (Maxwell & Pease, 1956). The CP is the principal source of the CSF (Lun *et al*, 2015; Spector *et al*, 2015).

We used a mouse model (Milner *et al*, 2015) similar to that of Feldner *et al* to test the differences between the permeability of the CP of $Mpdz^{+/+}$ and $Mpdz^{-/-}$ mice, and between the composition of their CSF, using approaches that have not been employed before to these ends. Based on our findings and detailed observations of the ultrastructure of the CP, we propose a new pathophysiological mechanism to explain the formation of hydrocephalus in the *Mpdz* LOF mouse model. The same mechanism could conceivably account for severe congenital hydrocephalus in humans carrying LOF variants of *MPDZ*.

# Results

## $Mpdz^{-/-}$ mice harbor severe congenital hydrocephalus

Out of a total of 112 mice bred by crossing heterozygous *Mpdz* mice, approximately 9% (10 mice) were homozygous for a gene-trap-induced mutation G510Vfs*19 (Milner *et al*, 2015). Consequently, the exons coding for PDZ domains 4–13 were truncated, likely resulting in nonsense-mediated mRNA decay. $Mpdz^{-/-}$ pups were indistinguishable from their littermates at birth, but their heads started to bulge and form a domed forehead as early as P4, becoming gradually more pronounced (Fig 1A). This is a malformation indicative of hydrocephalus. The lifespan of $Mpdz^{-/-}$ mice did not exceed 3 weeks, and by P18-P21, they were approximately 35% lighter than their wild-type littermates (Fig 1B). To substantiate the presence of hydrocephalus in the brains of $Mpdz^{-/-}$ mice, and to

distinguish between metabolically active and inert, possibly necrotic tissue, we imaged the brains of $Mpdz^{+/+}$ and $Mpdz^{-/-}$ mice by $^{18}$F-fluorodeoxyglucose positron emission tomography (PET). The images revealed low emission levels in most of the cranial volume of $Mpdz^{-/-}$ mice in comparison with the control $Mpdz^{+/+}$ mice, indicating low levels of metabolic activity (Fig 1C and D). We did not detect low PET emissions in the brain parenchyma of the $Mpdz^{-/-}$ mice, ruling out occurrence of necrosis tissue foci larger than 0.7 mm (Rodriguez-Villafuerte *et al*, 2014).

To elucidate the morphology of the brain and the ventricles of $Mpdz^{-/-}$ hydrocephalic mice, we analyzed brains of P18-P21 mice by magnetic resonance (MR) T2-weighted imaging. $Mpdz^{-/-}$ mice harbored CSF-filled lateral ventricles that coalesced into a vast single void (Fig 2A). The average total volume of $Mpdz^{-/-}$ ventricles was approximately 50-fold larger than the volume of the average total ventricle volume of $Mpdz^{+/+}$ mice (Fig 2B). The superior and the lateral cortices of $Mpdz^{-/-}$ mice were compressed by the enlarged lateral ventricles to a thickness of < 1 mm. Despite this acute deformation, the total volume of the brains of $Mpdz^{-/-}$ mice did not differ significantly from the volume of $Mpdz^{+/+}$ mice (Fig 2B), possibly because of the larger overall size of the brain. The MRI did not detect lesions in the brain parenchyma of $Mpdz^{-/-}$ mice. To date, all $Mpdz^{-/-}$ mice harbored severe hydrocephalus with little variation between individuals.

## MRI contrast medium leaks through the choroid plexus of $Mpdz^{-/-}$ mice

Gadolinium (Gd) chelate, a contrast medium used clinically to image the vascular system, does not normally cross the blood-brain or blood-CSF barriers (Breger *et al*, 1989). We reasoned, therefore, that degradation in the integrity of the CPEC monolayer could result in contrast medium penetration into the ventricles that would be detectable by T1-weighted imaging. The location of the lateral ventricles in mice with normal brains was identified in T2-weighted coronal brain images using hematoxylin–eosin (HE)-stained coronal sections as guide (Fig EV1A). We then measured the time course of the signal intensity at locations in the T1-weighted images matching the ventricles identified in the T2-weighted images (Fig EV1B). The time course of the signal intensity in the images of Gd-injected $Mpdz^{+/+}$ brains was irregular, lacking a recognizable temporal trend (Fig EV1C). Using again HE-stained coronal sections, we identified the CP villi attached to the top and sides of an elevated region at the bottom of the enlarged merged lateral ventricles in coronal T2-weighted images of $Mpdz^{-/-}$ mice (Fig 2C). This CP configuration is similar to the morphology of the CP in human hydrocephalic brains (Cardoza *et al*, 1988; Al-Dosari *et al*, 2013). Unlike the $Mpdz^{+/+}$ brains, we were able to identify visually the contrast medium signal in T1-weighted MR images of $Mpdz^{-/-}$ mouse brains (Fig 2D). The location of the signal in T1-weighted coronal images of $Mpdz^{-/-}$ brains corresponded accurately to the location of the CP in the T2-weighted images. The trend of the time course of the MR signal intensity sampled in the T1-weighted coronal images was unambiguously upward, peaking within the 10-min duration of the experiments (Fig 2E). These images indicate that the contrast medium leaked through the CP of $Mpdz^{-/-}$ mice into its abnormally enlarged and merged lateral ventricles.

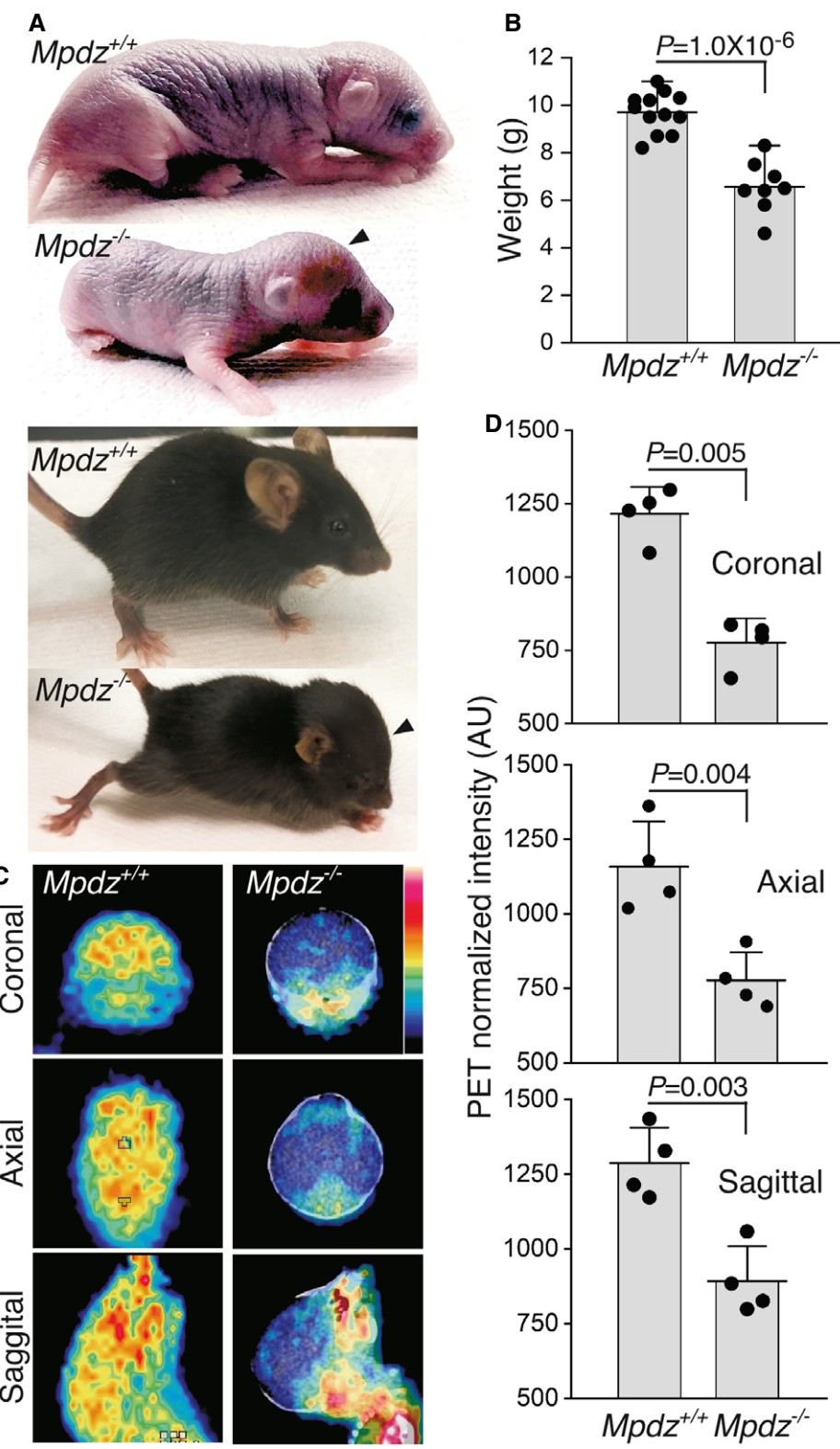

**Figure 1.  Hydrocephalus was detected by PET in *Mpdz*$^{-/-}$ mice.**

A  Images of P4 (top) and P21 (bottom) *Mpdz*$^{+/+}$ and *Mpdz*$^{-/-}$ mice. Arrowheads point to the domed foreheads of the latter.

B  Mean weights of P18-P21 *Mpdz*$^{+/+}$ and *Mpdz*$^{-/-}$ mice ($n$ = 8–12, mean ± SD; the value of $P$ was determined by two-tailed Student's $t$-test).

C  Coronal, axial, and sagittal (top to bottom) PET images of P18-P21 *Mpdz*$^{+/+}$ and *Mpdz*$^{-/-}$ mice. The emission intensity is shown as a 7-point temperature scale from black (0) to white (7).

D  Mean PET emission intensities in the indicated brain sections ($n$ = 4, mean ± SD; the values of $P$ were determined by two-tailed Student's $t$-test).

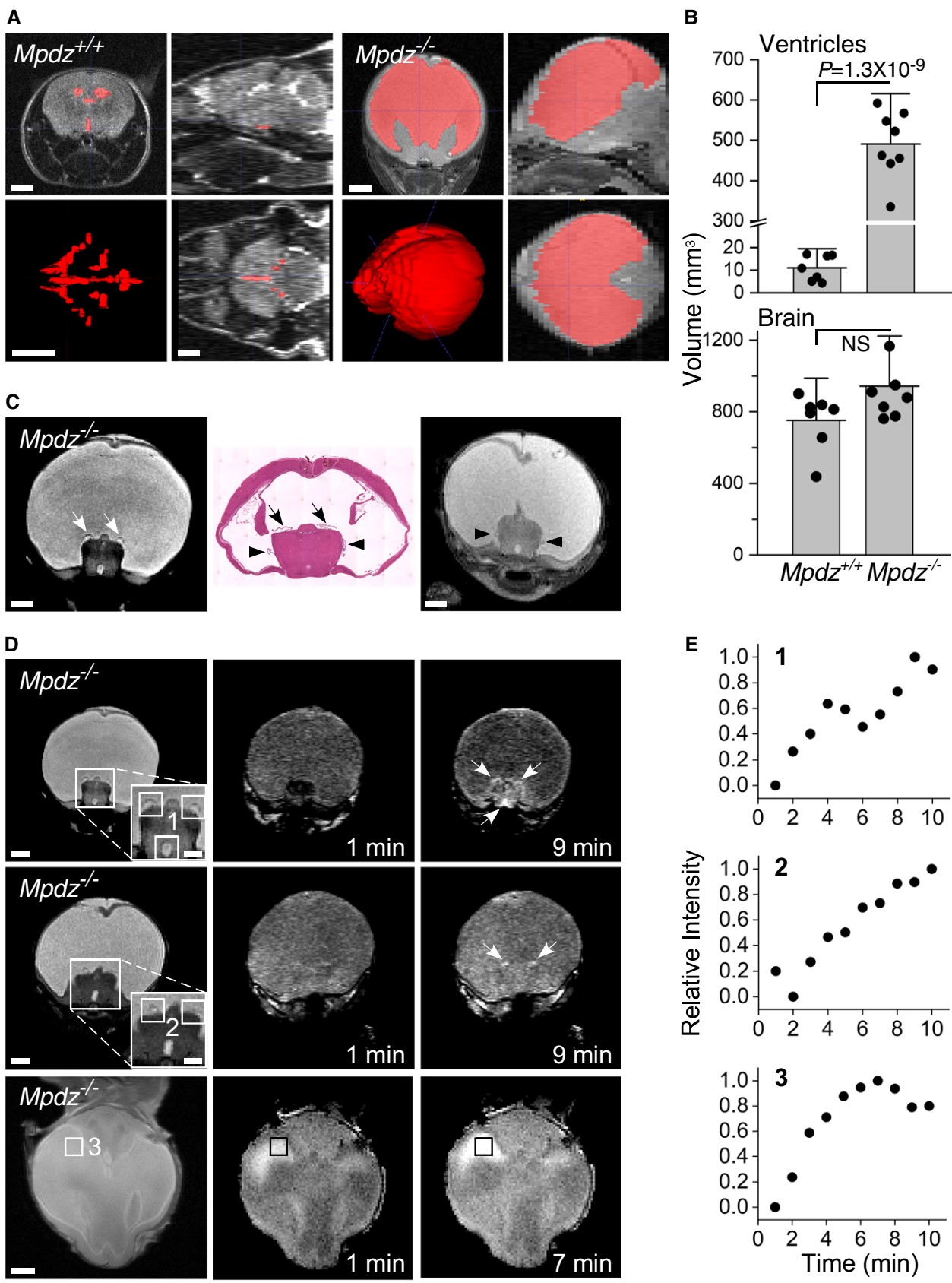

**Figure 2.**

◄

**Figure 2.  Severe hydrocephalus and leakage of contrast medium were detected in *Mpdz*⁻/⁻ mice by MRI.**

A  Coronal, sagittal, and axial (clockwise from top left corner image of each genotype) MR images of *Mpdz*⁺/⁺ and *Mpdz*⁻/⁻ P18-P21 mice. Ventricles are pseudo-colored in red and 3D-reconstructed in the lower left panels.

B  Means of total ventricle and brain volumes of *Mpdz*⁺/⁺ and *Mpdz*⁻/⁻ mice ($n$ = 8, mean ± SD; the values of $P$ were determined by two-tailed Student's $t$-test).

C  A coronal HE-stained section (center) flanked by anatomically corresponding coronal T2-weighted MR images of *Mpdz*⁻/⁻ P18-P21 mice. Arrows or arrowheads show the match between the lateral ventricle CP in the HE section and in the MR images.

D  Duplicate rows of T2-weighted coronal images and anatomically corresponding T1-weighted coronal images at 1 min and at the peak-signal time point after contrast medium injection; the bottom row shows a similar set of axial images. Areas surrounded by squares are magnified in the insets; arrows mark the contrast medium signal in the T1-weighted images. Each row corresponds to one mouse aged 18–21 days.

E  Time courses of the normalized T1-weighted image intensities corresponding to the areas surrounded by numbered squares on the T2-weighted images.

Data information: Scale bars, 1 mm; insets, 0.5 mm.

## The Sylvian aqueduct of the *Mpdz*⁻/⁻ mouse is stenotic

The MR-imaged *Mpdz*⁻/⁻ brain (Fig 2C) and the comparison of HE-stained sections of *Mpdz*⁺/⁺ (Fig 3A) and *Mpdz*⁻/⁻ (Fig 3B) brains indicated that the large cranial void in the brains of *Mpdz*⁻/⁻ mice resulted from the expansion and merger of the lateral ventricles, whereas the volume of the 3rd ventricle did not change noticeably. Fixed brains sections do not maintain the original dimensions of the organ. Though it was not evident that the Sylvian aqueduct is stenotic (Fig 3B), we injected Evans blue into the lateral brain ventricles of *Mpdz*⁺/⁺ and *Mpdz*⁻/⁻ mice. The aqueduct of the *Mpdz*⁻/⁻ mouse appeared stenosed in the ex vivo images of injected brain hemispheres (Fig 3C). Unlike the lateral ventricle or the aqueduct and fourth ventricle of the *Mpdz*⁺/⁺ mouse, little of the injected dye seeped into the surrounding parenchyma during the overnight incubation of the brain in fixative. This indicates that the flow through the aqueduct of the *Mpdz*⁻/⁻ mouse was slower than in the *Mpdz*⁺/⁺ mouse. Similar to *L1*⁻/⁻ mice (Rolf *et al*, 2001), aqueduct stenosis in *Mpdz*⁻/⁻ mouse could have been a result of the compression of the brain rather than a cause of hydrocephalus.

## Mpdz is localized apically in CPECs; its deficiency induces depletion of tight junction proteins

We probed the CPs of *Mpdz*⁺/⁺ and *Mpdz*⁻/⁻ mice with antibodies specific to Mpdz and to several junction transmembrane and membrane-associated proteins to identify potential structural differences between their respective CPEC junctions. Mpdz was detected exclusively near the apical face of CPECs from third ventricle villi (Fig 4A). This localization encompasses the sites of the tight junctions, which are the topmost structure in intercellular junctions. However, the conspicuous abundance of Mpdz on the CPEC apical surface suggests it may play an additional role unrelated to intercellular junction maintenance. The tight junction-associated scaffold protein ZO1 appeared more abundant in the CPEC monolayer of CP villi from the lateral ventricles of *Mpdz*⁺/⁺ mice (Fig 4B), whereas the abundances of the adherens junction protein epithelial (E)-cadherin were similar to each other (Fig 4C). Since quantification of the immunofluorescence of two-dimensional (2D) sections is not a robust measure of the overall abundance of the probed protein in the 3D CP, we opted to simulate the deficiency of Mpdz by knocking down *MPDZ* in human (h) primary CPECs by lentiviral transduction of *MPDZ*-targeted shRNA. We then used immunoblotting to compare the abundances of ZO1, Jam-C, and E-cadherin, to those of

hCPECs transduced by non-targeting shRNA. The protein abundances measured by densitometry of the immuno-adsorbed protein bands were similar to those suggested by the corresponding immunofluorescence images (Fig 4D), confirming that the abundances of ZO1 and Jam-C were lower in *Mpdz*⁻/⁻ mice or in hCPEC wherein *Mpdz* was knocked down, whereas that of E-cadherin did not change.

## Epithelial cells of *Mpdz*⁻/⁻ CP and their intercellular junctions are structurally and functionally defective

Transmission electron microscopy (TEM) detected substantial structural differences between the CPEC monolayers of *Mpdz*⁺/⁺ and *Mpdz*⁻/⁻ mice. While the structures of CP villi from the lateral ventricles of *Mpdz*⁺/⁺ and *Mpdz*⁻/⁻ mice imaged by TEM appeared grossly similar to each other (Fig 5A), examination at higher magnification uncovered the presence of a large number of voids of varying sizes in the CPECs of *Mpdz*⁻/⁻ mice (Fig 5B). Furthermore, the length of the adherens junctions was shorter, whereas the length of spaces between adjoining *Mpdz*⁻/⁻ CPECs was longer in comparison with adjoining *Mpdz*⁺/⁺ CPECs (Fig 5B). CPEC tight junctions, the foremost barrier to paracellular permeability (Zihni *et al*, 2016), between the CPECs of lateral ventricle villi from P18-P21 *Mpdz*⁻/⁻ mice were shorter, wider, and less electron-dense than those of *Mpdz*⁺/⁺ mice (Fig 5C), indicating that their protein concentration was lower than that of *Mpdz*⁺/⁺ CPEC junctions. TEM imaging of CPEC tight junctions of P12-P14 mice revealed similar differences, though they were subtler than at P18-P21 (Fig EV2). A second outstanding difference between *Mpdz*⁺/⁺ and *Mpdz*⁻/⁻ CPECs was the state of the mitochondria. Numerous mitochondria in CPECs from lateral ventricle villi of *Mpdz*⁻/⁻ mice lacked a major part of their cristae, and some of those contained autophagosomes (Fig 5D). It is possible that some of the voids in the *Mpdz*⁻/⁻ CPECs were remnants of fully dissolved mitochondria. The structural differences between the CPEC tight junctions in *Mpdz*⁺/⁺ and in *Mpdz*⁻/⁻ mice suggest that the former were less impervious to leakage than those between *Mpdz*⁺/⁺ CPECs, likely accounting for the contrast medium leakage from the CP into the lateral brain ventricles of *Mpdz*⁻/⁻ mice. Unlike CPECs, we did not detect structural differences between the junctions of the fenestrated endothelial cells that comprise the walls of the CP capillaries in *Mpdz*⁺/⁺ and *Mpdz*⁻/⁻ mice (Fig EV3).

To determine the dependence of the barrier function of CPEC monolayers on Mpdz, we compared the time course of impedance, an established permeability surrogate (Bischoff *et al*, 2016), of

**Figure 3. The Sylvian aqueduct of the *Mpdz*<sup>−/−</sup> mouse is stenotic.**

A  Coronal HE-stained brain sections of a P18 *Mpdz*<sup>+/+</sup> mouse (out of a total of three) showing the 3<sup>rd</sup> ventricle, and the proximal and distal sections of the aqueduct of Sylvius; these features are marked by arrows in the insets.

B  The same, for P18 *Mpdz*<sup>−/−</sup> mice.

C  Top views of brains of P7 *Mpdz*<sup>+/+</sup> and *Mpdz*<sup>−/−</sup> mice that show the sites of Evans blue injection, and the midline planes (dashed line) sectioned to produce the sagittal images below. They show the extent of Evans blue spread in the ventricles and surrounding tissue (one out of three experiments). In, injection; LV, lateral ventricle; SA, Sylvian aqueduct; V3, third ventricle; V4, fourth ventricle.

Data information: Scale bars, 1 mm; insets, 0.25 mm.

human papilloma CPEC (hpCPEC) monolayers (Ishiwata *et al*, 2005; Feldner *et al*, 2017) transduced by *MPDZ*-targeting or by non-targeting shRNA. The impedance of the control group was persistently higher throughout the 70-h duration of the measurement (Fig 5E). While the MPDZ-deficient hpCPECs (Fig 5E, inset) reached a plateau in approximately 60 h, the impedance of the control group

of cells continued to rise, reaching a 42 % higher amplitude than the MPDZ-deficient hpCPECs. The lower impedance of these cells indicates they posed lower resistance to the alternating electrical current passing through the monolayer they formed, compared to the control group, and are, therefore, more permeable, in agreement with the structural findings.

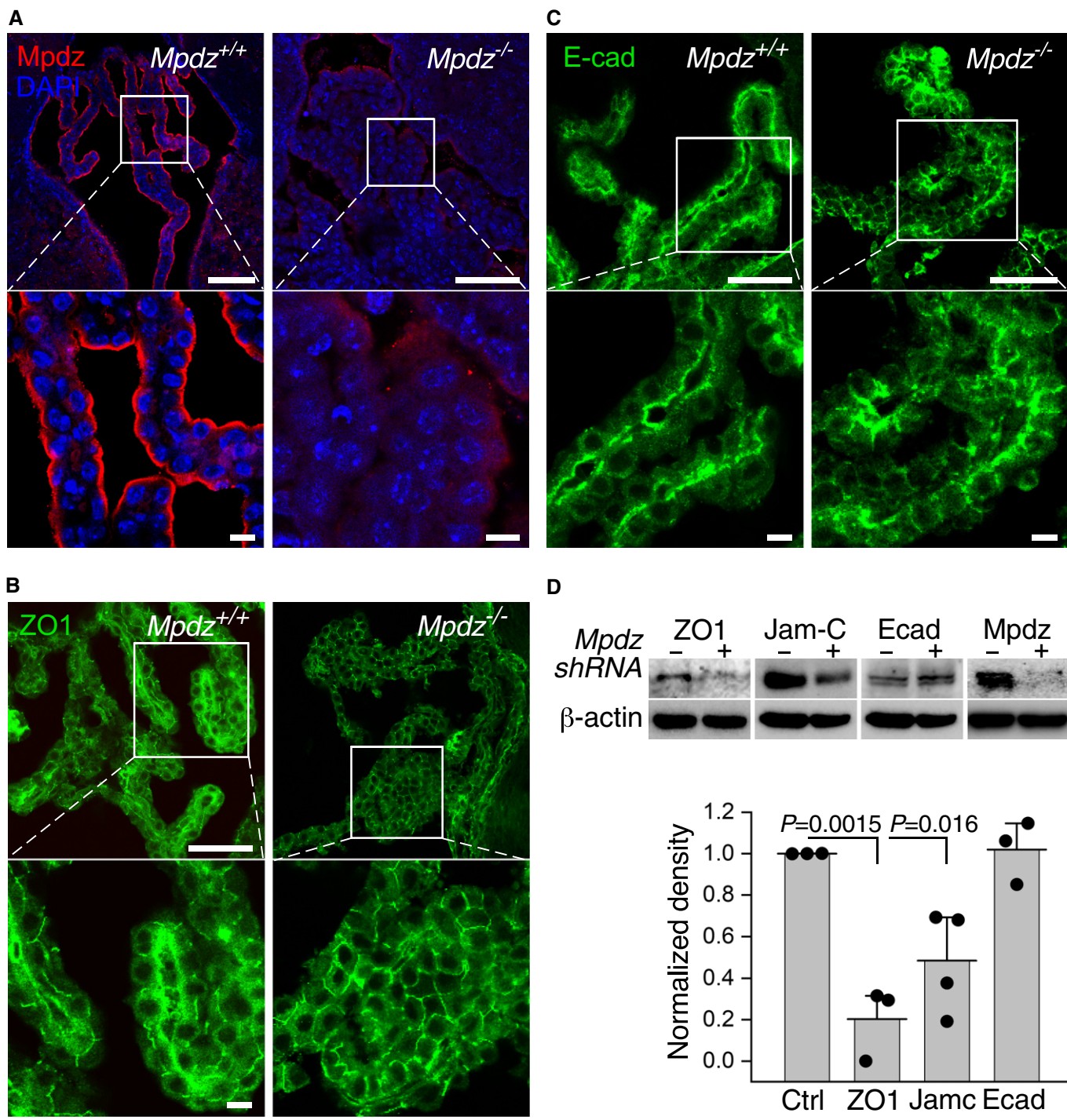

**Figure 4.  Mpdz was localized proximal to the apical surface of CPECs.**

A    Immunofluorescence images of 10-μm-thick sections of CP from third ventricle villi of *Mpdz*^+/+ and *Mpdz*^−/− mice were immunolabeled as shown. The areas in the square frames are magnified in the bottom panels. Scale bars, top panels, 100 μm; bottom panels, 10 μm.

B, C    CP sections from lateral ventricle villi immunolabeled for ZO1 (B) or E-cadherin (E-cad; C). The images are representative of two *Mpdz*^+/+ mice and three *Mpdz*^−/− mice. Scale bars, top panels, 100 μm; bottom panels, 10 μm.

D    Immunoblots with the indicated antibodies (top) and their quantifications. The densitometry measurements were normalized relative to the signal of the samples transduced by non-targeting shRNA (Ctrl). Note that the same β-actin immunoblot was used twice because the Jam-C and E-cadherin samples were immunoblotted on the same membrane (mean ± SD, *n* = 3; the values of *P* were determined by two-tailed Student's *t*-test).

Source data are available online for this figure.

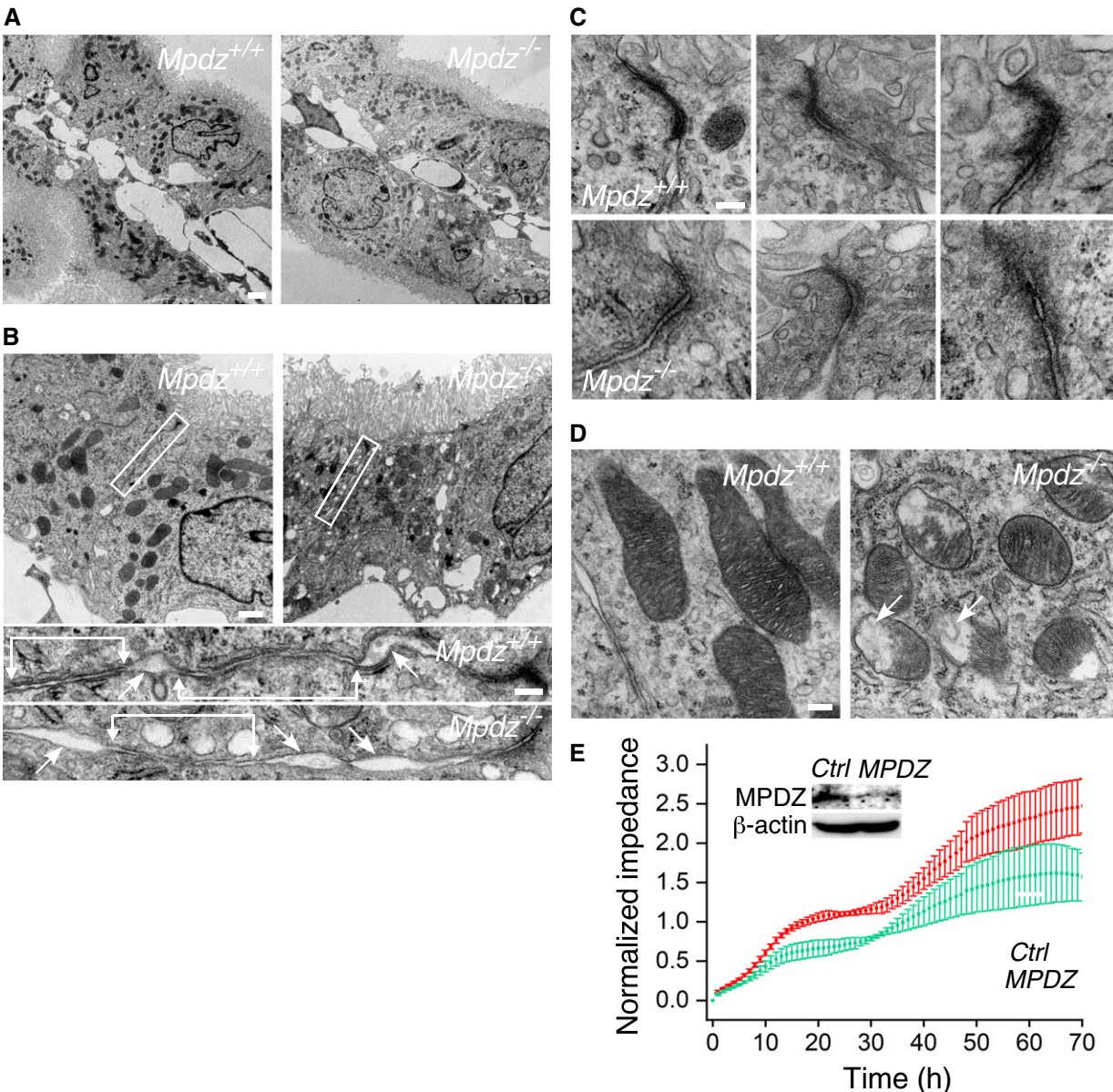

**Figure 5.  CPECs of *Mpdz*$^{-/-}$ mice harbored structural defects.**

A   TEM images of longitudinal sections of lateral ventricle CP villi of P18-P21 *Mpdz*$^{+/+}$ and *Mpdz*$^{-/-}$ mice. Scale bar, 2 μm.

B   A large number of voids of varying sizes were evident in the CPECs of *Mpdz*$^{-/-}$ mice. Higher magnification images of the areas in the rectangles are shown underneath the panels. Adherens junctions are denoted by horizontal lines and arrows; voids are indicated by arrows. Scale bar, 1 μm; insets, 200 nm.

C   Triplicate images of tight junctions proximal to the apical faces of CPECs in *Mpdz*$^{+/+}$ and *Mpdz*$^{-/-}$ mice. Scale bar, 100 nm.

D   Mitochondria were smaller and frequently lacked large portions of their cristae. Autophagosomes are present in some of them (arrows). Scale bar, 200 nm. The images are representative of two *Mpdz*$^{+/+}$ mice and two *Mpdz*$^{-/-}$ mice.

E   Time course and standard deviations of the impedance of confluent hpCPEC monolayers that were transduced by either *MPDZ* or non-targeting (Ctrl) shRNA. Each record represents four wells (mean ± SD). MPDZ immunoblot of each cell group is shown in the inset.

Source data are available online for this figure.

## Transcytosis through the CP is higher in *Mpdz*$^{-/-}$ than in *Mpdz*$^{+/+}$ mice

To fully characterize the barrier function of the CP, we compared the transcellular permeabilities of the CP of *Mpdz*$^{+/+}$ and *Mpdz*$^{-/-}$ mice by tracking fluid-phase uptake and receptor-dependent transcytosis. We used horseradish peroxidase injected in vivo for ex vivo chromogenesis by hydrogen peroxide-induced oxidation of 3,3′-diaminobenzidine (DAB) (Broadwell & Brightman, 1983). The number of internalized DAB particles per cell in CPEC sections from lateral ventricle villi of *Mpdz*$^{-/-}$ mice was approximately sixfold higher than in those of *Mpdz*$^{+/+}$ mice (Fig 6A), suggesting that the

rate of fluid-phase uptake through the CPECs of $Mpdz^{-/-}$ mice was substantially higher than its rate in $Mpdz^{+/+}$ mice. Quantification of the number of DAB particles per cell showed that the fluid-phase uptake by CPECs of $Mpdz^{-/-}$ mice was more than sixfold higher relative to that of $Mpdz^{+/+}$ mice (Fig 6B). Many of the larger endocytosed DAB particles consisted of concentric layers

(magnified fields in Fig 6A), a phenomenon seen in previous studies on ependymal cells (Broadwell & Sofroniew, 1993) and brain endothelial cells (Broadwell *et al*, 1996). The endocytosed DAB formed a morphologically heterogenous population of particles in the CPECs of both mouse genotypes, most of which were located in macropinosomes, following engulfment by ruffles on

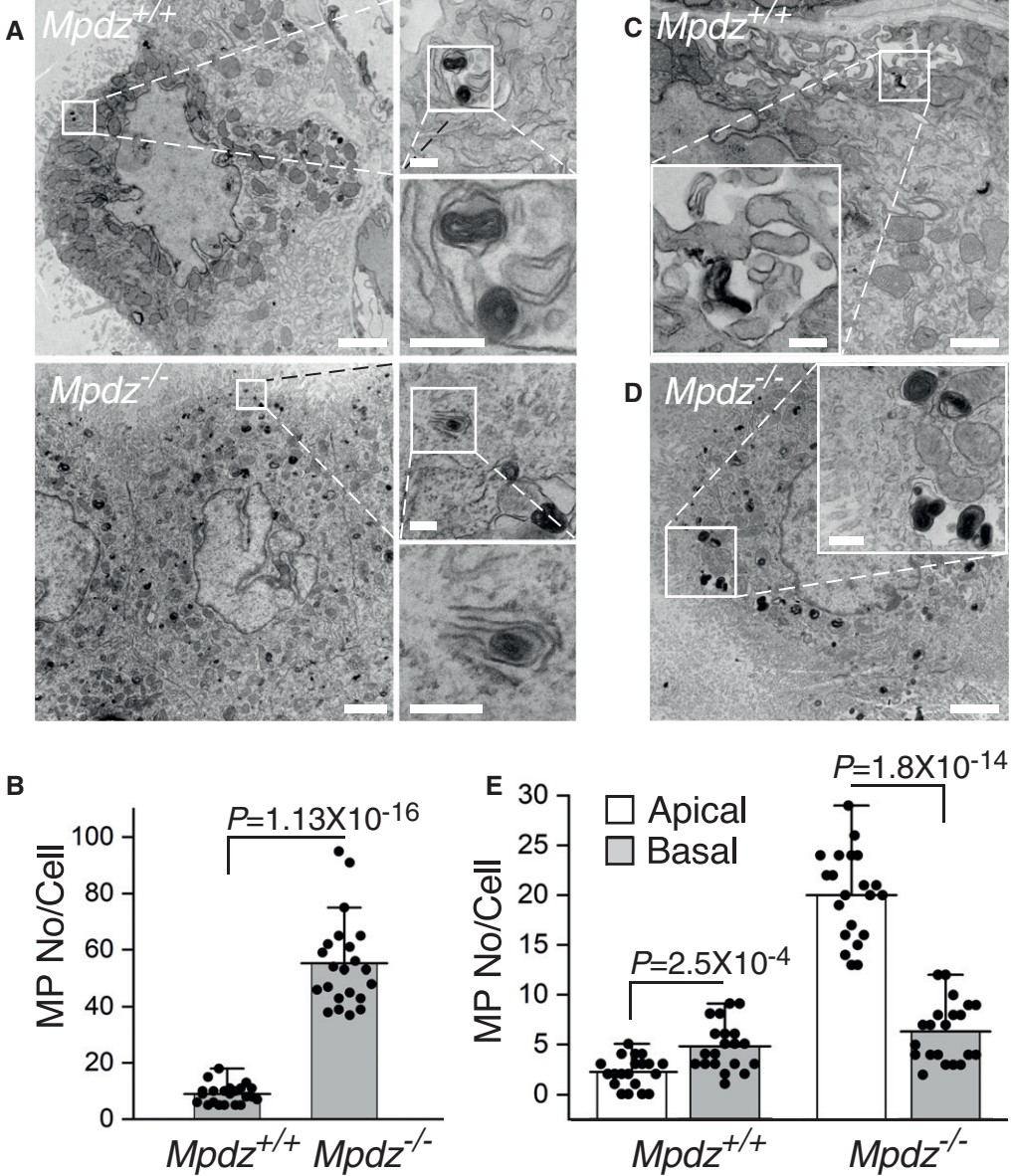

**Figure 6. Fluid-phase uptake by CPECs is higher in $Mpdz^{-/-}$ relative to $Mpdz^{+/+}$ mice.**

A TEM images of CPEC sections from lateral ventricle villi of two $Mpdz^{+/+}$ and two $Mpdz^{-/-}$ P14-P16 mice injected with HRP. The CPs were reacted with hydrogen peroxide and DAB ex vivo. The dark particles are DAB deposits internalized by micropinocytosis. The magnified fields to the right show individual particles engulfed in macropinosomes. Note the layered structure of the particles, and the macropinosome that is open to the ventricular space in the $Mpdz^{-/-}$ section. Scale bars, 1 μm; insets, 100 nm.

B Mean number of engulfed DAB particles per cell in $Mpdz^{+/+}$ and $Mpdz^{-/-}$ mice (mean ± SD, n = 22; the value of P was determined by two-tailed Student's t-test).

C A CPEC section showing the engulfment of a DAB particle by the cell's basal ruffles in the magnified field. Scale bars, 1 μm; insets, 200 nm.

D A CPEC section showing a preponderance of macropinosomes close to the apical face of the cell, and a magnified field that contains several macropinosomes. Scale bars, 1 μm; insets, 200 nm.

E Mean numbers of DAB-containing macropinosomes close to the apical or basal sides of CPECs from $Mpdz^{+/+}$ or $Mpdz^{-/-}$ mice (mean ± SD, n = 20; the values of P were determined by two-tailed Student's t-test).

the CPEC basal surface (Fig 6C). This is similar to previous observations on DAB endocytosis in mouse CPECs (Balin & Broadwell, 1988). We noticed a marked difference between the distributions of the macropinosomes in CPECS of $Mpdz^{+/+}$ and $Mpdz^{-/-}$ mice: Whereas in the former the majority was located near the basal side facing the lumen of the CP villus (Fig 6C), most macropinosomes were located near the apical side of the CPECs of the latter, facing the ventricle (Fig 6D). In CPECs of $Mpdz^{+/+}$ mice, more than twice macropinosomes were near the basal than near the apical side; in CEPCs of $Mpdz^{-/-}$ mice, the ratio was reversed to more than threefold macropinosomes near the apical side (Fig 6D). This difference between macropinosome distributions indicates that the rate of macropinosome transcytosis from the CPEC basal to apical side was higher in lateral ventricle villi of $Mpdz^{-/-}$ than of $Mpdz^{+/+}$ mice.

To compare receptor-mediated transcytosis through the CPECs of $Mpdz^{+/+}$ and $Mpdz^{-/-}$ mice, we focused on the abundance and endocytosis of the low-density lipoprotein (LDL) receptor, because apolipoprotein E (ApoE), an LDL carrier, was the most over-abundant protein in the CSF of $Mpdz^{-/-}$ mice (see below). Since LDL traverses the blood-brain barrier by LDLR-mediated endocytosis, followed by transcytosis (Dehouck et al, 1997), we tested the presence of LDLR in CPECs of $Mpdz^{+/+}$ and $Mpdz^{-/-}$ mice. The higher intensity of the immunofluorescence signal emanating from CP sections from lateral ventricle villi of $Mpdz^{-/-}$ mice indicated that LDLR was more abundant than in its $Mpdz^{+/+}$ counterpart (Fig 7A). The mean fluorescence intensity per CPEC of $Mpdz^{-/-}$ mice was higher by approximately 40% (Fig 7B). The higher transcytosis through the $Mpdz^{-/-}$ CP could have been a response to the condition of hydrocephalus that is unrelated to the $Mpdz$ LOF or could have been induced by the absence of functional Mpdz. To test the causal connection between $MPDZ$ expression and LDLR abundance, we knocked down $MPDZ$ in hCPECs by lentiviral transduction of $MPDZ$-targeting shRNA and compared the abundances of LDLR to hCPECs transduced by non-targeting shRNA. LDLR was more abundant in the MPDZ-deficient hCPECs by more than twofold, suggesting that the depletion of MPDZ induced an increase in the amount of LDLR through a yet unknown pathway (Fig 7C). To test the correlation between the abundance of MPDZ and the extent of LDLR transcytosis, we tracked the constitutive endocytosis (Zou & Ting, 2011) of endogenous LDLR in hCPECs that were transduced by $MPDZ$-targeting or by non-targeting shRNA. LDLR was close to twofold more abundant on the cell surface prior to the initiation of endocytosis and higher by 54% after 8 min of constitutive endocytosis in MPDZ-deficient hCPECs (Fig 7D).

**Protein concentration is substantially higher in the CSF of $Mpdz^{-/-}$ mice**

The total protein concentration in the CSF of $Mpdz^{-/-}$ mice at P18-P21 was more than twice higher than that of $Mpdz^{+/+}$ mice (Fig 8A). We compared the composition of the serum and the CSF of $Mpdz^{+/+}$ and $Mpdz^{-/-}$ mice by tandem liquid chromatography and mass spectroscopy (LC-MS/MS). The serum compositions of the two genotypes were highly similar, with only one over-abundant protein in the CSF of either $Mpdz^{+/+}$ or $Mpdz^{-/-}$ mice (Fig 8B). In contrast, the composition of the CSF differed substantially between the two genotypes (Fig 8C). We detected a total of 313 proteins in all the samples pooled together, after excluding hemoglobin subunits α and β, catalase, peroxiredoxin, and carbonic anhydrase-1 as serum contaminants (You et al, 2005) (Dataset EV1). Out of these, all but 13 proteins had been detected in murine CSF in previous studies (Cunningham et al, 2013; Smith et al, 2014; Dislich et al, 2015). The CSF of $Mpdz^{-/-}$ mice contained 23 proteins that were either absent in the CSF of $Mpdz^{+/+}$ mice or that were at least twofold significantly more abundant in the CSF of $Mpdz^{-/-}$ mice (Table 1). Only two proteins were at least twofold significantly more abundant in the CSF of $Mpdz^{+/+}$ mice. The similarity between the serum protein compositions of $Mpdz^{+/+}$ and $Mpdz^{-/-}$ mice shows that protein over-abundance in the CSF of $Mpdz^{-/-}$ mice is not a direct result of the serum composition but a consequence of the functional differences between the CPs of $Mpdz^{+/+}$ and $Mpdz^{-/-}$ Mice.

All the proteins that were at least twofold more abundant in the CSF of $Mpdz^{-/-}$ mice are secreted or known to have soluble forms (fibronectin, gelsolin, and vitronectin). They can be classified into several molecular function groups: components of the blood coagulation cascade (complement C4-B, complement factor H, properdin/complement factor P, and the fibrinogen β chain), extracellular matrix (extracellular matrix protein 1 (Ecm1), fibronectin, and vitronectin), lipoproteins (apolipoproteins D and E), immune response (beta-2-microglobulin and the macrophage colony-stimulating factor 1 receptor), cytokines and cytokine-binding proteins (α-fetoprotein/insulin-like growth factor-binding protein 1 (Afp), α-2-HS-glycoprotein, granulin (Grn), hepatocyte growth factor activator, and insulin-like growth factor (IGF)-binding proteins 2 and 4), enzymes and enzyme-binding proteins (α-2-macroglobulin-P, lysozyme C-2, and sulfhydryl oxidase 1), and protease inhibitors (angiotensinogen and pigment epithelium-derived factor). Out of these, Afp, Ecm1, and Grn were found only in the CSF of $Mpdz^{-/-}$ mice.

The over-abundance of proteins in the CSF of $Mpdz^{-/-}$ mice could have resulted from an overall increase in the rate of CSF production. To test this premise, we compared the abundances of the $Na^{+}$-$K^{+}$-$2Cl^{-}$ cotransporter 1 (Nkcc1), recently shown to contribute significantly to CSF production (Steffensen et al, 2018), in the CP of $Mpdz^{+/+}$ and $Mpdz^{-/-}$ mice. Quantification of confocal images of CP sections indicated that Nkcc1 was approximately 75% more abundant in the CP from lateral ventricle villi of $Mpdz^{-/-}$ mice (Fig 8D). Similar to LDLR, knockdown of $MPDZ$ in hCPECs was accompanied by an increase of approximately 68% in NKCC1 abundance (Fig 8E).

The presence of blood coagulation cascade proteins in the CSF has been linked to neuro-inflammation (Ehling et al, 2011; Wang et al, 2011). ApoE is produced in the CP and secreted in response to neuronal injury (Lehtimaki et al, 1995). Its presence may indicate that the pressure exerted by the expanding hydrocephalus injured the brain parenchyma. The four IGF-binding proteins are inhibitors of the insulin growth factor-like receptor (Srinivas et al, 1993; Kelley et al, 2002). Their over-abundance could be a negative feedback triggered by the swelling of the $Mpdz^{-/-}$ brain, to suppress further IGF1-induced neurogenesis (Annenkov, 2009). Angiotensinogen is the precursor protein of angiotensins, generated by cleavage of its N-terminus by renin (Skeggs et al, 1957). The angiotensins maintain blood pressure homeostasis in response to fluid intake or loss (Gardes et al, 1982). Their production could have been triggered by the elevated brain blood pressure in hydrocephalic mice. Pigment epithelium-derived factor supports neuron differentiation and growth (Steele et al, 1993). The functional classifications of

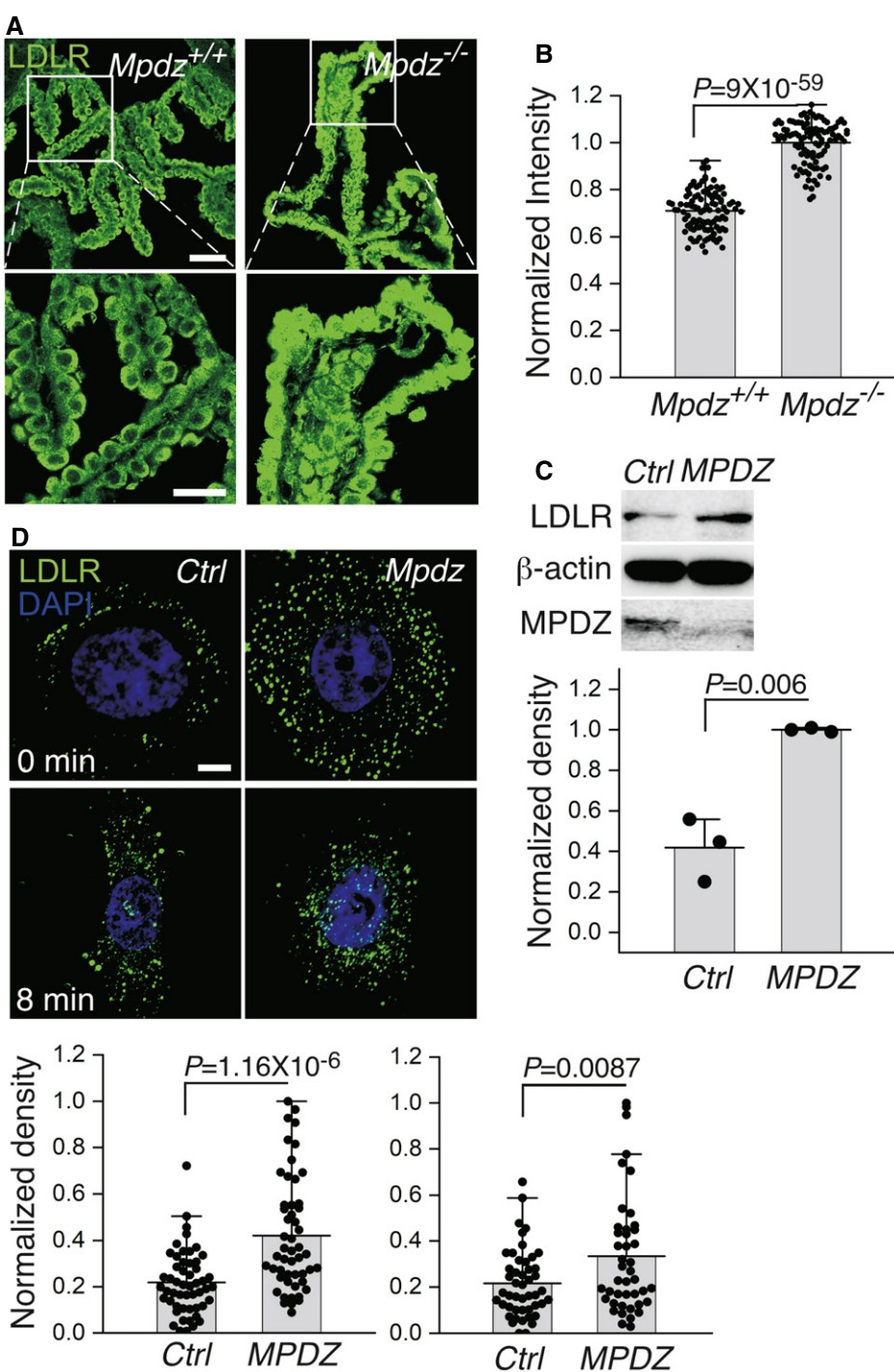

**Figure 7.  LDLR abundance and endocytosis by CPECs are higher in *Mpdz*⁻/⁻ relative to *Mpdz*⁺/⁺ mice.**

A   Immunofluorescence images of 10-μm-thick sections from lateral ventricle CP villi of *Mpdz*⁺/⁺ and *Mpdz*⁻/⁻ P14-P16 mice immunolabeled by anti-LDLR. Scale bars, 50 (top) and 25 (bottom) μm.

B   Mean fluorescence intensities (normalized relative to the highest recorded intensity) per cell in several LDLR-immunolabeled CP sections (mean ± SD, *n* = 101; the value of *P* was determined by two-tailed Student's *t*-test).

C   Immunoblots showing the abundances of LDLR and MPDZ in hCPECs transduced by *MPDZ*-targeting or non-targeting shRNA. Mean abundances were quantified by densitometry of the LDLR bands, normalized relative to the β-actin bands (mean ± SD, *n* = 3; the values of *P* were determined by two-tailed Student's *t*-test).

D   Fluorescence images of hCPECs transduced by either *MPDZ* or non-targeting (Ctrl) shRNA and immunolabeled by anti-LDLR either before (0 min) or after 8 min of constitutive endocytosis of LDLR. Scale bar, 10 μm. The histograms below show the mean fluorescence intensities per hCPEC in each cell group, normalized relative to the highest recorded intensity (mean ± SD, *n* = 41–53; the values of *P* were determined by two-tailed Student's *t*-test).

Source data are available online for this figure.

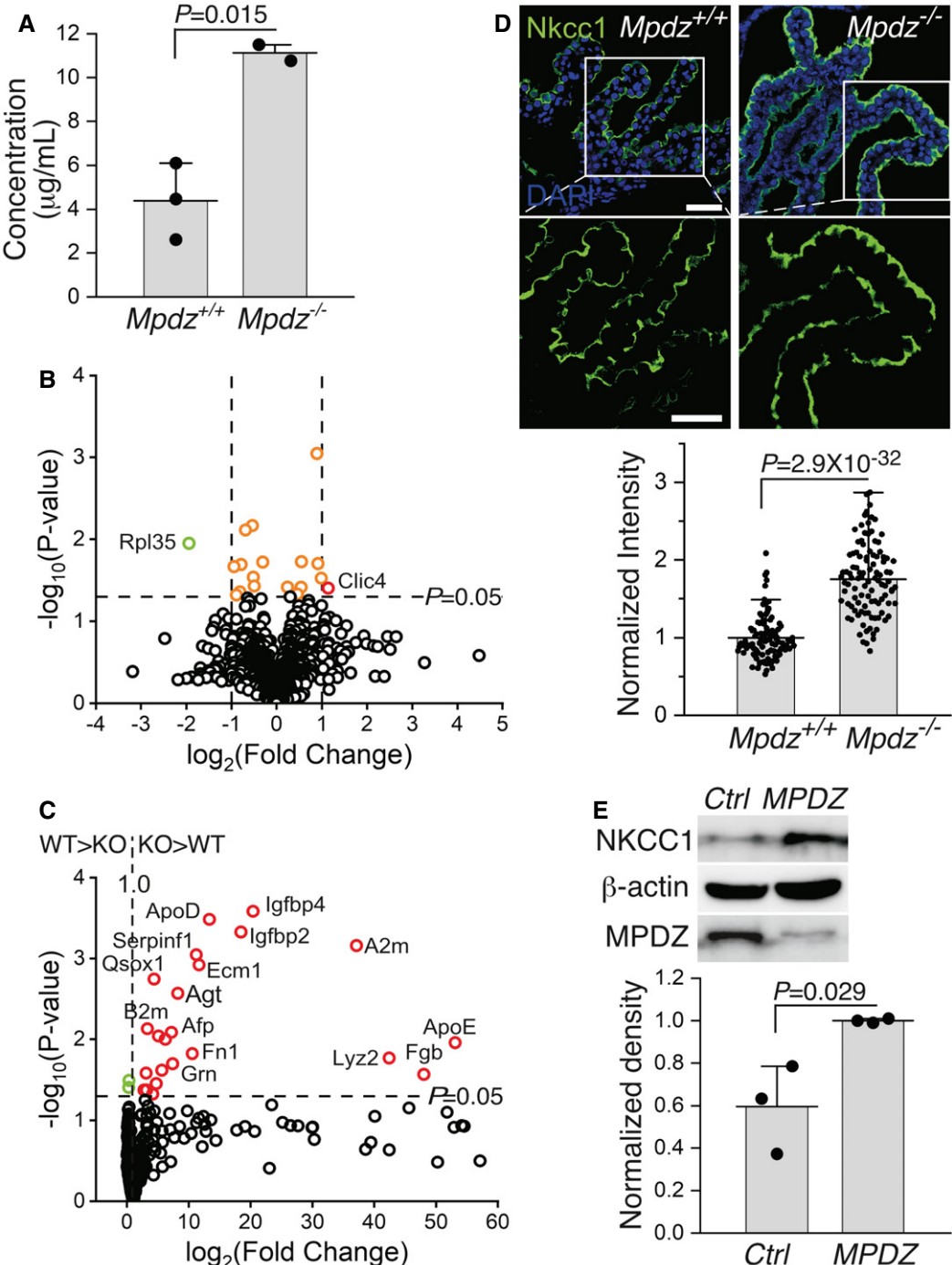

**Figure 8. The protein content in the CSF of *Mpdz*$^{-/-}$ mice was higher than in the CSF of *Mpdz*$^{+/+}$ mice.**

A    Mean protein concentration in the CSF of *Mpdz*$^{+/+}$ and *Mpdz*$^{-/-}$ P14-P21 mice (mean ± SD, *n* = 2–3; the values of *P* were determined by two-tailed Student's *t*-test).

B    Volcano plot showing that the protein contents in sera of P15-P21 *Mpdz*$^{+/+}$ and *Mpdz*$^{-/-}$ mice are similar to each other.

C    A volcano plot showing that 23 proteins are at least twofold significantly more abundant in the CSF of P15-21 *Mpdz*$^{-/-}$ mice (red circles, protein names are indicated), whereas only two proteins were more abundant in the CSF of *Mpdz*$^{+/+}$ mice (green circles). The CSF composition was analyzed in three *Mpdz*$^{-/-}$ and three *Mpdz*$^{+/+}$ mice.

D    Images of 10-μm-thick sections from lateral ventricle CP villi of *Mpdz*$^{+/+}$ and *Mpdz*$^{-/-}$ mice, immunolabeled for Nkcc1. The areas in the marked squares are magnified below. Scale bars, top, 50 μm, bottom 20 μm. The histograms show the mean fluorescence intensities per hCPEC in each cell group, normalized relative to the highest recorded intensity (mean ± SD, *n* = 103; the value of *P* was determined by two-tailed Student's *t*-test).

E    Immunoblots of LDLR and Mpdz, with β-actin as loading control. The MPDZ immunoblot shows the efficiency of the shRNA-mediated knockdown. Bands from three immunoblots were quantified by normalization to β-actin bands (mean ± SD; the value of *P* was determined by two-tailed Student's *t*-test).

Source data are available online for this figure.

**Table 1.  Over-abundant proteins in the CSF of *Mpdz*$^{-/-}$ mice, and their parameters in the LC-MS/MS analysis.**

| Protein name | Gene ID | MW (kDa) | Isoelectric point | Average LFQ intensities (×10$^3$) | | LFQ KO/WT ratio | *P*-value |
| --- | --- | --- | --- | --- | --- | --- | --- |
| | | | | *Mpdz*$^{+/+}$ | *Mpdz*$^{-/-}$ | | |
| Extracellular matrix protein 1 | Ecm1 | 48.356 | 6.28 | 9.247 | 108.1 | 11.691 | 0.0012 |
| Fibronectin | Fn1 | 272.53 | 5.3 | 594.234 | 6292.53 | 10.589 | 0.015 |
| Gelsolin | Gsn | 85.94 | 5.72 | 2186.3 | 9651.9 | 4.415 | 0.0018 |
| Vitronectin | Vtn | 54.84 | 5.56 | 409.711 | 1280.37 | 3.125 | 0.026 |
| Complement C4-B | C4b | 192.91 | 8.7 | 586.938 | 3005.67 | 5.121 | 0.0091 |
| Complement factor H | Cfh | 139.14 | 6.54 | 3049.323 | 9739.17 | 3.194 | 0.042 |
| Properdin | Cfp | 50.32 | 7.43 | 101.668 | 283.75 | 2.791 | 0.042 |
| Fibrinogen β chain | Fgb | 54.75 | 8.32 | 45.285 | 2175.04 | 48.03 | 0.027 |
| Apolipoprotein D | ApoD | 21.53 | 5.46 | 45.159 | 604.07 | 13.377 | 0.0003 |
| Apolipoprotein E | ApoE | 35.87 | 5.46 | 955.539 | 50725 | 53.085 | 0.011 |
| β-2-microglobulin | B2m | 13.78 | 7.97 | 425.157 | 1419.5 | 3.339 | 0.0074 |
| Macrophage colony-stimulating factor 1 receptor | Csf1r | 109.18 | 5.87 | 22.71 | 141.25 | 6.22 | 0.01 |
| α-fetoprotein | Afp | 67.34 | 7.97 | 78.845 | 567.28 | 7.195 | 0.008 |
| α-2-HS-glycoprotein | Ahsg | 37.32 | 5.94 | 7831.367 | 44357.3 | 5.664 | 0.024 |
| Granulin | Grn | 63.46 | 6.41 | 13.719 | 101.49 | 7.398 | 0.020 |
| Hepatocyte growth factor activator | Hgfac | 70.57 | 11.45 | 26.246 | 109.89 | 4.187 | 0.047 |
| Insulin-like growth factor-binding protein 2 | Igfbp2 | 32.846 | 7.2 | 62.722 | 1156.4 | 18.437 | 0.0005 |
| Insulin-like growth factor-binding protein 4 | Igfbp4 | 27.81 | 6.62 | 2.61 | 53257 | 20.381 | 0.0003 |
| α-2-macroglobulin-P | A2m | 164.35 | 6.1 | 125.699 | 4668.7 | 37.142 | 0.0007 |
| Lysozyme C-2 | Lyz2 | 16.69 | 8.99 | 49.978 | 2118.97 | 42.398 | 0.017 |
| Sulfhydryl oxidase 1 | Qsox1 | 63.34 | 6.43 | 60.672 | 288.44 | 4.754 | 0.035 |
| Angiotensinogen | Agt | 51.99 | 5.18 | 106.615 | 877.37 | 8.229 | 0.003 |
| Pigment epithelium-derived factor | Serpinf1 | 46.23 | 6.45 | 36.352 | 408.52 | 11.238 | 0.0009 |

Proteins are grouped functionally into seven sets (from top to bottom): extracellular matrix, blood coagulation cascade, lipoproteins, immune response, cytokines and cytokine-binding proteins, enzymes and enzyme-binding proteins, and protease inhibitors. LFQ, label-free quantification. The values of *P* were determined by two-tailed Student's *t*-test.

these proteins suggest that their over-abundance is part of a multifaceted physiological response to the stress imposed on the brain by the expanding hydrocephalus, rather than a haphazard collection of unrelated proteins. We did not detect lesions in the brain of hydrocephalus-harboring P21 mice either macro- or microscopically. Correspondingly, none of the over-abundant proteins in the CSF of *Mpdz*$^{-/-}$ mice is cytoskeletal; gelsolin has a secreted isoform (Yin *et al*, 1984). Furthermore, there was no statistically significant difference between the concentrations of serum albumin, amyloid A-4, amyloid P-component, and paraoxonase-1 in the CSF of *Mpdz*$^{+/+}$ and *Mpdz*$^{-/-}$ mice (Dataset EV1). It is unlikely, therefore, that the over-abundance of these proteins reflects breakdown of the brain parenchyma.

## Discussion

Our results indicate that *Mpdz* LOF has a twofold effect on the barrier function of murine CP: It increases both passive paracellular permeability, as indicated by the leakage of the GD-based MRI contrast medium, and the rate of protein transcytosis, as indicated by the higher abundance of close to 10% of all the proteins found in the CSF of *Mpdz*$^{+/+}$ and *Mpdz*$^{-/-}$ mice. The first effect is likely caused by degradation of the integrity of CPEC intercellular junctions, because we did not find structural differences between the junctions of the fenestrated endothelial cells of CP vessels. The second effect, which is likely to be a physiological response to the stress inflicted on the brain by the formation of hydrocephalus, could involve both cell monolayers.

The first mouse model of hydrocephalus, generated by a *L1cam* LOF mutation, harbored neural defects, but the ependymal cells that coat the ventricle lumen appeared normal (Dahme *et al*, 1997). The brain malformations typical to the L1 syndrome were attributed to the inter-neuronal adhesion function of L1CAM (Miura *et al*, 1992), but no mechanism had been invoked to account specifically for the formation of hydrocephalus. *AP1S2* is also located in the X chromosome and, like *L1CAM*, underlies several syndromic defects, one of which is hydrocephalus. In some cases, *AP1S2*-linked

hydrocephalus was associated with stenosis of the Sylvian aqueduct (Saillour *et al*, 2007). No causal connection has been made between *AP1S2* LOF mutations and the formation of hydrocephalus.

The hydrocephalus observed in carriers of *CCDC88C* mutations resulted presumably from a dysfunction of Wnt non-canonical signaling, because these mutations truncate the C-terminus of DAPLE, the protein it encodes, disabling its binding to the PDZ domain of DVL. Since DVL is required for the formation of Wnt-induced planar polarity of ependymal cells (Ohata *et al*, 2014), it is conceivable that *CCDC88C*-linked hydrocephalus was caused by the loss of the alignment of the cilia of the ependymal cells on the lumen of the ventricles and the resulting slowdown of CSF flow. Furthermore, DAPLE binds to (Redwine *et al*, 2017) and functions as an activating adaptor of dynein (Reck-Peterson *et al*, 2018), a molecular motor that is required for cilium motility (Gibbons & Rowe, 1965). To date, the premise that *CCDC88C* mutations cause hydrocephalus by impeding ciliary function has not been tested.

The localization of MPDZ to epithelial and endothelial intercellular junctions, its binding of multiple transmembrane junction proteins, and its high abundance in the CP are collectively suggestive of a causal connection between the formation of hydrocephalus and *MPDZ* LOF mutations in humans and mice. Presumably, if MPDZ is required for the stabilization of transmembrane proteins at endothelial and epithelial cell junctions, its absence would impair junction integrity. Though this scenario had been entertained for both human (Al-Dosari *et al*, 2013; Saugier-Veber *et al*, 2017) and murine (Feldner *et al*, 2017) *MPDZ*-linked hydrocephalus, no direct causative connection has been established between possible *MPDZ* LOF deleterious effects on intercellular junction integrity and the formation of hydrocephalus in humans or mice carrying *MPDZ* LOF mutations. The first reported *MPDZ* mutation linked to severe congenital hydrocephalus in humans would have truncated 12 of the 13 PDZ domains (Al-Dosari *et al*, 2013). A subsequent study reported three new mutations that caused truncation of MPDZ within PDZ domain #3, a frameshift within PDZ domain #1, and a truncation within PDZ domain #5 (Saugier-Veber *et al*, 2017). All three mutations were expected to introduce a premature stop codon and result in nonsense-mediated decay of the transcript. In all three affected individuals, the aqueduct of Sylvius was stenotic, and the ependyma in the aqueduct as well as in the third and fourth ventricles linked by the aqueduct was interspersed with denuded focal lesions. Since MPDZ is abundant in aqueduct ependymal cells of normal individuals, the ependymal lesions were attributed to the formation of defective tight junctions, but there was no direct evidence for this eventuality.

The initiation of hydrocephalus in the *Mpdz* LOF mouse model of Feldner *et al* was attributed to deterioration of ependymal cell integrity as a result of weakened tight junctions, similar to the explanation of *MPDZ*-linked human hydrocephalus. No evidence was provided to support this premise. In contrast, we provided direct evidence that the CP of *Mpdz*$^{-/-}$ mice was leaky and structurally defective, implying that the hydrocephalus formed because of imbalance between the rate of CSF production and removal from the lateral ventricles through the relatively narrow 3$^{rd}$ ventricle, aqueduct, and 4$^{th}$ ventricle, resulting in the expansion and merging of the lateral ventricles into the large void we detected in the brains of *Mpdz*$^{-/-}$ mice. Our pathophysiological model is not necessarily in conflict with the findings of Feldner *et al* but would consider the

ependymal lesions and stenosis of the Sylvian aqueduct observed by them as possible effects rather than causes.

The composition of the CSF of *Mpdz*$^{-/-}$ mice is indicative of the stress posed on the brain by the swelling of the lateral ventricles. This response could have been instigated, conceivably, by the ependymal lesions observed by Feldner *et al*, as well as by other injuries to the brain parenchyma that became ultimately fatal at 3 weeks of age. The protein over-abundance in the CSF of *Mpdz*$^{-/-}$ mice is likely the result of augmented transcytosis, a process that occurs normally in the CP (Grapp *et al*, 2013). The higher rate of fluid-phase uptake by the CPECs of *Mpdz*$^{-/-}$ mice could have contributed to fluid accumulation in their lateral ventricles. The large imbalance between the protein contents in the CSF and the interior of the choroid plexus could have caused an osmotic pressure gradient between the two compartments, further driving fluid flow from the choroid plexus into the ventricles. At this time, we are aware of a single comparative proteomic analysis of CSF of normal and hydrocephalic subjects (Finehout *et al*, 2004). This study did not analyze, however, CSF from patients harboring congenital hydrocephalus, but that of a single subject with normal pressure hydrocephalus and from two normal patients. The study detected a relatively low total number of only 82 proteins, possibly because the samples were resolved by 2D electrophoresis rather than undergoing liquid chromatography prior to mass spectrometry. This may have reduced the sensitivity of the assay.

The manner, in which the *Mpdz* loss-of-function mutation in vivo or *Mpdz* knockdown in vitro brings about an increase in the abundances of LDLR, and Nkcc1, and a decrease in the abundances of ZO1 and Jam-C, is unknown at this time.

## Materials and Methods

### Animals

Mice were housed in ventilated cages and supervised by the Thomas Jefferson University Laboratory Animal Services Animal (Welfare Assurance Number D16-00051) and used according to protocols approved by the Institutional Animal Care & Use Committee and renewed annually. Thomas Jefferson University is accredited by the Association for Assessment and Accreditation of Laboratory Animal Care. C57BL/6J *Mpdz*$^{+/-}$ mice (Milner *et al*, 2015) were kindly gifted by Dr. Kari Buck, Oregon Health and Science University. The *Mpdz* mutation was generated by Bay Genomics as gene-trap model MpdzGt(XG734)Byg. It translates to a G510Vfs*19 change in the protein sequence. *Mpdz*$^{-/-}$ mice were bred by heterozygote crossing and used between P0 and P21. Within each genotype, the inclusion of gender and of individual mice was random. The reporting of in vivo experiments in this article conforms with the Animal Research: Reporting of In Vivo Experiments (ARRIVE) guidelines (Kilkenny *et al*, 2010).

### Cell culture, *MPDZ* knockdown, and immunoblotting

Human hCPEC (ScienCell Research Laboratories) were grown in Epithelial Cell Medium purchased from the same provider. *Mpdz*

was knocked down by transduction with lentivirus expressing a validated shRNA (clone TRCN0000349430, MilliporeSigma) or a non-targeting shRNA (pLKO.1-puro shRNA Control, MilliporeSigma). The virus was produced in HEK293 cells (American Type Cell Culture) that were transfected (Fugene HD, Promega) with packaging and coat protein plasmids (pCMVR8.74 and pMD2.G, respectively, Addgene). Lentivirus particles were concentrated from the cell culture medium by centrifugation (Speedy Lentivirus Purification kit, ABM). hCPEC were lysed 3 days after lentivirus transduction in RIPA buffer (Thermo Fisher Scientific (TFS)) supplemented with protease inhibitor cocktail tablets (Roche), and immunoblotted with LDLR antibody (MilliporeSigma) diluted 1:1,000. LDLR band density was quantified by an ImageJ plugin and normalized relative to the load-control band density (antibody to β-actin, MilliporeSigma diluted 1:2,000).

### PET imaging

Mice aged 18–21 days were injected through the tail vein with 0.4–0.5 mCi of [$^{18}$F]fluorodeoxyglucose ($^{18}$FDG, Siemens Medical Solutions USA, Inc) and imaged as described (Seidler *et al*, 2006). Emission intensity was quantified by measuring the signal per unit area in two-dimensional regions of interest (ROIs), after subtracting the background signal.

### Magnetic resonance imaging

Mice aged 18–21 days anesthetized by 3% isoflurane inhalation were imaged with a 4.7 Tesla, 50-cm horizontal bore instrument by T1- and T2-weighted modalities at the Small Animal Imaging Facility of the University of Pennsylvania. Body core temperature was kept at 37°C by a rectal probe. Mice were injected through the tail vein with 50 μl 0.1 mmol/kg diethylenetriaminepentaacetic acid (DPTA) dimeglumine salt Gd chelate (M$_r$ 938 Da, Magnevist®, Bayer) immediately after the completion of T2-weighted imaging, and underwent T1-weighted imaging for 10 min. Ventricle and total brain volumes were quantified by the ITK-SNAP (V3.2) application (http://www.itksnap.org) (Yushkevich & Gerig, 2017). Signal intensities in T1-weighted images were quantified at each time point by ImageJ (version 1.52d, https://imagej.nih.gov) at ROIs corresponding to the location of the ventricles in T2-weighted images of the same section. Change in total intensity relative to the initial value was normalized in each ROI by division by the difference between the maximum and minimum values.

### Histology, immunohistochemistry, and optical microscopy

Anesthetized mice aged 18-21 were perfused intracardially with physiological saline, followed by 4% paraformaldehyde (PFA) in PBS, pH 7.4. Brains were removed, fixed overnight in 4% PFA/PBS, immersed in 30% (w/v) sucrose in TBS, pH 7.4, at 4°C until they became submerged, and frozen to 180°C in an optimal cutting temperature medium (Fisher Scientific). Coronal sections were cut at 20 μm (HM560, Microm), mounted on glass slides (SuperFrost Plus, TFS), and air-dried overnight. Sections were blocked in 5% normal goat serum for 1 h at 23°C and incubated overnight at 4°C by primary antibodies to ZO1 (Invitrogen) diluted 1:200, LDLR

(eBiosciences) diluted 1:100, E-cadherin (BD Biosciences) diluted 1:200, Jam-C (R&D Systems) diluted 1:200, Nkcc1 (Cell Signaling Tech.) diluted 1:200, and MPDZ [gift of Prof. Elior Peles, Weizmann Institute of Science (Poliak *et al*, 2002)] diluted 1:100. The primary antibodies were detected by secondary antibodies to the IgG of the appropriate host species of the primary antibodies, conjugated to Alexa Fluor® 488 or 555 (TFS) diluted 1:1,000. The sections were mounted in anti-fade 4′,6-diamidino-2-phenylindole (DAPI)-containing medium (ProLong, TFS). Immunofluorescence intensity was quantified with ImageJ by measuring the average pixel intensity within manually defined contours of 2D cell images, to exclude effects of cell size. Only cells that were sectioned through the nucleus were quantified. HE-stained sections were imaged by scanning with a 10× objective (EVOS, FSI) and stitching the fields. Immunolabeled sections were imaged by laser-scanning confocal microscopy (Nikon A1R+).

### Transcytosis

Mice aged 2–3 weeks were anesthetized and injected with HRP (0.5 mg/g body weight HRP type II, Aldrich) in PBS through the left ventricle. After 5-min circulation, mice were euthanized, and the brain was excised and immersed immediately in 4% PFA/PBS during CP dissection. The dissected CP was fixed by 5% glutaraldehyde and 4% PFA in 0.1 M sodiumcacodylate for 1 h at 23°C, followed by overnight fixation by 4% PFA in 0.1 M sodiumcacodylate at 4°C. After fixation, the tissue was washed three times and immersed overnight in 0.1 M sodiumcacodylate. CPs were then incubated in 0.01% hydrogen peroxide and 0.5 mg/ml DAB (Sigma-Aldrich) in 0.05 M Tris–HCl pH 7.6 for 45 min at 23°C. The CPs were stored in 5% glutaraldehyde and 4% PFA in 0.1 M sodiumcacodylate until preparation for TEM. Transcytosis was quantified by counting the number of DAB-stained vesicles per cell.

### LDLR endocytosis

Human papilloma CPECs grown on glass coverslips were incubated on ice with antibody to LDLR (eBiosciences) diluted 1:100, for 30 min, washed by ice-cold growth medium, and transferred to 37°C growth medium. After 8 min, the coverslips were washed by ice-cold PBS pH 2.5 for 30 s to remove remaining cell surface antibody, fixed with 4% PFA/PBS for 20 min @ RT, and permeabilized with 0.1% Triton X-100 in PBS for 5 min at 23°C. The primary antibody was detected by a 30-min incubation at 23°C with anti-chicken IgG conjugated to Alexa Flour 488 (TFS). Coverslips were mounted on slides (ProLong, TFS) before imaging by confocal microscopy (Nikon A1R+).

### Evans blue intracranial injection

Mice anesthetized by ketamine/xylazine were injected through the left ventricle with five microliter Evans blue dye diluted 1:100 in PBS by a 10-ml syringe (Hamilton). The syringe was left in the injection site to prevent fluid reflux during the next 5 min, after which the mice were euthanized by decapitation. The heads were immediately fixed in 4% paraformaldehyde/PBS overnight. Brains were dissected and imaged on a stereomicroscope (Leica MZ10F).

## Electron microscopy

Brains removed as described were dissected under a stereomicroscope (Leica MZ10F) to isolate the choroid plexus from the lateral ventricles. Isolated choroid plexi were fixed overnight in 2% glutaraldehyde, stained with 1% OsO4 and 0.5% uranyl acetate, pelleted in 2% agarose (MilliporeSigma, Type IX ultra-low gelling temperature), dehydrated in an acetone/water dilution series, and embedded in Araldite resin (EMbed 812, Electron Microscopy Sciences). Sections of 60 nm cut by ultramicrotome (Leica UCT) were mounted on square-mesh or oval-hole grids and imaged by TEM (FEI Tecnai G12) at 80 keV.

## Mass spectrometry and measurement of CSF protein concentrations

Mice aged 18–21 days were anesthetized as described, their heads shaved, and immobilized stereotactically (Robot Stereotaxic, NeuroStar) while maintaining body temperature at 37°C (TCAT-2LV controller, Physitemp). Anesthetic depth was verified by assessing reflexes before and during surgery. After peeling off the skin to expose the cranium, a 1.0-mm-diameter hole was drilled above the left lateral ventricle at −0.1 mm relative to the bregma, −0.80 mm from the midline, and 2.5 mm beneath the dura. A volume of 3–5 µl CSF was drawn from normal mice by a 10-µl syringe (Hamilton) at a rate of 1 µl/min and stored immediately on dry ice. CSF was drawn identically from hydrocephalic mice aside from the location of the hole that was drilled 2 mm left of the peak of the skull. A volume of 2 µl of each sample was reconstituted in 50 µl 6 M urea/100 mM Tris, pH 8.0 and digested by 3 µg trypsin overnight at room temperature. Samples were desalted by spin columns (Pierce™ C18, TFS) and reconstituted in 30 µl 1% acetic acid. Volumes of 50 µl of each sample were injected into a reversed-phase capillary chromatography column (Acclaim™ PepMap™ C18, Dionex), eluted by acetonitrile/0.1% formic acid gradient at a flow rate of 0.3 µl/min into the intake of a linear trap quadrupole (Orbitrap Elite™, TFS) hybrid mass spectrometer, and electro-sprayed at 1.9 kV. Amino acid sequences were determined from peptide molecular weights and ion collision-induced dissociation spectra by searching the UniProt mouse protein database with MaxQuant (V1.5.2.8) application. CSF protein concentrations were measured by spectrometry of a 1:10 diluted CSF sample (NanoDrop, TFS).

## Statistics

The significance of the difference between means was determined by two-tailed Student's t-test. The null hypothesis was considered untrue if the probability satisfied the condition $P \leq 0.05$. Based on initial sample variance, we increased sample size to test the statistical significance of inter-group difference, to fulfill the above criterion. The variances were similar between the tested groups, except for the control groups in densitometry measurements, which were set to 1. We excluded outliers if they were 1.5× the interquartile difference (between the medians of the upper and lower halves of the dataset) above the top median or below to bottom median (Tukey's fence). This applied only to Fig 7D. The normality of all datasets was confirmed by the Shapiro–Wilk test (Shapiro & Wilk, 1965),

### The paper explained

#### Problem

Congenital hydrocephalus is a potentially life-threatening condition that occurs at a frequency of one case per 1,000 births. Its main characteristic is accumulation of cerebrospinal fluid in the chambers of the brain, leading to swelling of the brain that is confined by the rigid skull. The only current treatment is shunting, an invasive procedure that fails within 2 years in 50% of the cases. Though this condition is hereditary, the manner by which a mutation results in hydrocephalus is not understood.

#### Results

Magnetic resonance was used to image contrast medium in 3-week-old mice deficient of Mpdz, a large scaffold protein shared by humans. The medium leaked from the choroid plexus into the ventricles of the brain, showing that Mpdz is required for the integrity and function of the choroid plexus. The leakiness was caused by defects in the epithelial cell layer that surrounds the blood vessels of the choroid plexus. Another consequence of the loss of Mpdz was a more than twice higher protein concentration in the cerebrospinal fluid of the mutant mice.

#### Impact

Our study provides a straightforward connection between the mutation and the appearance of congenital hydrocephalus. The new insights may facilitate the development of non-invasive approaches for the treatment of the condition.

implemented by the Origin application (OriginLab). Sample identity was masked during data quantification of immunofluorescence images or of immunoblot bands. Blinding of investigators in regard to animals was not possible because the hydrocephalic mice were conspicuous.

# Data availability

The dataset of proteins identified in the CSF of $Mpdz^{+/+}$ and $Mpdz^{-/-}$ mice and their ratios is available in the PRIDE archive as PXD011535 (http://www.ebi.ac.uk/pride/archive/projects/PXD011535).

**Expanded View** for this article is available online.

## Acknowledgements

We thank Dr. Kari Buck, Oregon Health & Science University, for sharing C57BL/6J $Mpdz^{+/-}$ mice; Dr. Didier Trono, Lausanne Federal Polytechnic School, Switzerland, for sharing the pCMVR8.74 and pMD2.G plasmids; Dr. Gyorgy Csordas and Ms. Bodil Toma, Thomas Jefferson University, for assistance with TEM imaging; Prof. Elior Peles, Weizmann Institute of Science, Israel, for sharing a polyclonal antibody to MPDZ; Mr. Biao Zou and the University of Pennsylvania's Electron Microscopy Resource Laboratory for preparing and mounting tissue thin sections for TEM; Drs. Yi Fan and Menggui Huang, University of Pennsylvania's Department of Radiation Oncology, for help with MRI protocols; and Prof. Annie Laquerriere, University of Rouen, France, for sharing tissue samples. This study was supported by the Cardeza Foundation for Hematologic Research (JY, CS, RK, IS, and AH), by the National Heart Lung and Blood Institute (R01 grant HL119984 to AH), by the HHS | NIH | National Cancer Institute (NCI) (S10 grants OD012406 and RR23709 to MT; P30 CA56036

to the Bioimaging Shared Resource of the Sidney Kimmel Cancer Center, Thomas Jefferson University), and by the National Institutes of Health Office of the Director (S10 grant OD023436 to BW).

## Author contributions

JY and CS designed and performed in vitro experiments; RK prepared mice for MRI and functional in vivo experiments, and isolated CPs; LO performed immunofluorescence experiments; MDB harvested CSF and performed functional in vivo experiments; ZN performed TEM imaging; IS identified the first hydrocephalus-harboring mouse and managed the mouse cohorts; ST and FP-K performed PET imaging, and MT supervised them and analyzed data; EL and PS-V analyzed data and wrote the manuscript; BW supervised mass-spectrometry experiments and analyzed data; SP performed MRI and analyzed data; HI and HS contributed human papilloma choroid plexus epithelial cells; RS supervised the experiments performed by LO and MDB, analyzed data, and wrote the manuscript; AH initiated the study, organized its performance, designed experiments, prepared mice for MRI, isolated CPs, analyzed data, and wrote the manuscript.

## Conflict of interest

The authors declare that they have no conflict of interest.

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
