## [Review Process File · EMBO Molecular Medicine]

Murine *MPDZ*-Linked Hydrocephalus is Caused by Hyperpermeability of the Choroid Plexus

Junning Yang, Claire Simonneau, Robert Kilker, Laura Oakley, Matthew D. Byrne, Zuzana Nichtova, Ioana Stefanescu, Fnu Pardeep-Kumar, Sushil Tripathi, Eric Londin, Pascale Saugier-Weber, Belinda Willard, Mathew Thakur, Stephen Pickup, Hiroshi Ishikawa, Horst Schrotten, Richard Smeyne, and Arie Horowitz

Review timeline:

Submission date:	11 July 2018
Editorial Decision:	8 August 2018
Revision received:	25 September 2018
Editorial Decision:	24 October 2018
Revision received:	6 November 2018
Accepted:	9 November 2018

Editor: Céline Carret/Lise Roth

Transaction Report:

1st Editorial Decision

8 August 2018

Thank you for the submission of your manuscript to EMBO Molecular Medicine. I have now had the time to read your manuscript and the comments from the 2 referees whom we asked to evaluate your manuscript.

As you will see from the reports below, both referees acknowledge the potential interest of the study but also agree that substantial revisions will be needed to fully support the conclusions. Therefore, addressing the reviewers' concerns in full will be necessary for further considering the manuscript in our journal. EMBO Molecular Medicine encourages a single round of revision only and therefore, acceptance or rejection of the manuscript will depend on the completeness of your responses included in the next, final version of the manuscript.

Please also contact us as soon as possible if similar work is published elsewhere. If other work is published, we may not be able to extend the revision period beyond three months.

We look forward to receiving your revised manuscript.

***** Reviewer's comments *****

Referee #1 (Comments on Novelty/Model System for Author):

The analyses need to be performed at much earlier time points before overt damage through the intracerebral pressure occurs.

Referee #1 (Remarks for Author):

This manuscript addresses the pathogenesis of congenital hydrocephalus in *Mpdz*^{-/-} mice. The authors nicely demonstrate that MRI contrast agents diffuse into cerebrospinal fluid (CSF) in *-/-* mice but not in *+/+* controls. Also, the authors detect a higher protein content in CSF. Based on this they argue that hydrocephalus occurs by increased paracellular permeability at the choroid plexus. In contrast to a previous report in this journal they did not detect stenosis of the aqueduct. This is a highly interesting study. *Mpdz* has originally been described to be strongly expressed in epithelial cells of the choroid plexus. These cells form a tight barrier between blood plasma and CSF. Given the literature reports that *Mpdz* stabilizes cell-cell junctions, it is intriguing to conclude that hydrocephalus might be caused by increased flux of proteins and water across the blood-CSF barrier. As such, the pathogenesis would be through over-production of CSF. Little is known about this, but choroid plexus tumors can indeed cause hydrocephalus.

The manuscript needs very substantial revision to demonstrate that overproduction of CSF is the primary cause.

1) A major problem with this study is that only mice with very pronounced hydrocephalus were analyzed. The authors report that mice did not live longer than 3 weeks and they analyzed 18 to 21-day old mice. In this stage the intracranial pressure is obviously excessively high. One cannot rule out that the subsequent damage on brain tissue (actually indicated by decreased metabolic activity in Fig. 1) and potentially also blood-brain-barrier and blood-CSF allow penetration of contrast agent and proteins into CSF irrespective of the primary cause. Therefore, the authors should provide more information when the hydrocephalus is first detectable and repeat the tracer experiments at these early time points.

Can the authors measure the rate of CSF production in these mice?

The same problem with EM data in Figure 7. This is really intriguing and supports the authors' hypothesis. However, such permeability changes need to be demonstrated at much earlier time points.

2) Given the complexity in the *in vivo* setting the authors should try to demonstrate that deletion of *Mpdz* increases permeability of choroid plexus epithelial cells *in vitro*.

3) The H&E staining suggests an open aqueduct of Sylvius. However, more functional analysis of CSF flux through the ventricular system would be needed to exclude stenosis.

4) Figure 6: Why has only ZO-1 been analyzed? *Mpdz* binds to several TJ and AJ proteins and the literature suggests that loss of *Mpdz* does not lead to disassembly of junctions but rather to weakening. A detailed expression analysis of such proteins is needed otherwise such strong statements cannot be made.

5). Figure 8: Data on the LDLR are preliminary. A functional assay should be performed to show that loss of *MPDZ* affects LDLR-mediated endocytosis.

Minor comments:

Figure 3/4/5: Please, state the age of the animals and number of animals analyzed.

Figure 6/7: Number of biological replicates should be added.

Referee #2 (Comments on Novelty/Model System for Author):

The authors dissect the pathogenic mechanics of the *MPDZ* knockout mouse model of congenital human hydrocephalus. They neatly demonstrated the aberrant contrast enhancement in this model, suggesting a change in permeability of a structure in the ventricular system. Arguing for both an increase in the structure/permeability and an increase in secretory function of the choroid plexus, they then suggest ultrastructural changes in the choroid plexus and describe the proteomic

differences in the CSF composition of this hydrocephalus model. Overall, this would provide an important contribution to understanding the pathogenesis of hydrocephalus, but the mechanism is likely not as "straightforward" as described in the abstract. The choroid plexus may be involved in addition to other structures. The current interpretation of the results requires further experiments to support the conclusion. Major points to address include:

Major concerns:

1) Evidence to support altered transcytosis is not sufficient. EM images (TEM = transmission EM, not transmitted EM), may suffer from fixation problems, which commonly include formation of vacuoles. Quantification, sample numbers, ventricle location, and statistics are needed to make statements about junction length, for example. However as is, with the current approaches used and representative images shown, it is not possible to conclude that transcytosis is altered in this tissue. Classic approaches to examine transcytosis have not been used (e.g. Evans blue, HRP injections). Chow & Gu (Neuron 2017) provides one example on how some of these studies could be performed.

2) Extensive evidence in the field demonstrates that the CSF is not a serum filtrate, and this is clear from any number of fine, contemporary reviews on the topic. In addition, given the novelty of proteomic CSF analysis in hydrocephalus, we have no reference point of comparison to understand the described characterizations and make a clear interpretation. How unique is the altered proteome to this model of hydrocephalus? Would hydrocephalus by any other mechanism also lead indirectly to heightened CSF ApoE and other protein expression (via elevated intracranial pressure or ventriculomegaly/distension, etc)? It seems the CSF is sampled after puncturing the lateral ventricle. This injury alone may induce changes in the CSF. Why not use a more conventional approach vetted by many groups for obtaining pure samples of CSF (e.g. DeMattos et al., J Neurochem 2002)?

3) Which choroid plexus is used in analyses? The lateral ventricles are the most severely affected, but most images shown seem to represent morphology of the 3rd ventricle choroid plexus. The age of sample collections may also influence interpretation of results. It appears that samples are typically collected and analyzed around P18, but by this age, the hydrocephalus is extensive and the authors state that the mice are not viable past ~P21 (3 weeks). It seems that hydrocephalus that is this extensive and induces severe injury in the brain and its vasculature, would alter the interstitial fluid and CSF. Hence, it is unclear if the effects described in the present study are primary, secondary, tertiary....including earlier pre-symptomatic or early hydrocephalic samples to the study would be informative.

4) What is the anatomical source of the contrast enhancement in Figure 4? The data suggests the choroid plexus might be a source, but the data presented shows a somewhat asymmetric contrast enhancement pattern (bottom of Figure 4B) and in other cases appears either highly focal and also in the aqueduct (top of Figure 4B). Given the presumed broad expression pattern of Mpdz, it is possible that alternative sources would include ependyma, circumventricular organs, or other potential areas of blood-brain barrier breakdown and vascular defects. Contrast enhancement might also suggest hemorrhage, which is not supported by the proteomic studies, but it would be worthwhile noting by presenting additional data regarding comparisons in content of basic serum proteins or albumin or by showing evidence for preserved vascularity of the choroid plexus. Lastly, what happens to contrast enhancement after 10min- does it dissipate in the CSF or persist in particular regions?

5) Regarding the paracellular permeability mechanism and the presence of 23 new proteins in the CSF, what are the protein size and electrostatic charge comparisons? Do these proteins appear selected based on any criteria that could be more easily explained by ventricular permeability alone? What does the silver stain look like, and what is the typical CSF protein concentration in these samples?

6) It is unclear why the authors think the Feldner 2017 Mpdz knockout mouse has aqueductal stenosis, in contrast to the Milner 2015 mouse model? The discrepancy suggests that some of the data may be unique to the Milner knockout model?

Minor concerns

- 1) Why was PET imaging of the mice performed? There is no experimental justification for it as a starting experiment, and it does not offer any additional data that was not presented in the MRI data. This should be clarified or the data moved to supplemental information.
- 2) Figure 3- what was the n number of mice?
- 3) Figure 8 legend does not mention LDL-R
- 4) Fig 3C mislabeled T2 and T1 in the legend. Fig4C legend also mislabeled T2 (should write T1)
- 5) "weighted" is misspelled on page 7 and in the legend of Fig 4

In its current form, I regret that I cannot support publication of this study as a short report in EMBO Molecular Medicine.

1st Revision - authors' response

25 September 2018

Responses to the Referees' comments

I thank the Referees for their constructive and insightful comments, for their promptness, and for their recognition of the significance and interesting nature of our study. The manuscript (MS) was revised by adding substantial new data from experiments we performed in response to the Referees' comments, culminating in 15 new figure panels. The text was rewritten where necessary, including the Discussion, either in response to the Referees' comments or in view of the new data. The text was revised also to mention the recent article of Fury et al., which reported four new hydrocephalus-linked genes (Neuron vol. 99, pp. 1–13, July 2018). The new text was underlined to distinguish it from preexisting text.

Referee 1

1. *The authors should provide more information when the hydrocephalus is first detectable and repeat the tracer experiments at these early time points. A major problem with this study is that only mice with very pronounced hydrocephalus were analyzed. The authors report that mice did not live longer than 3 weeks and they analyzed 18 to 21-days old mice. In this stage the intracranial is obviously excessively high. One cannot rule out that the subsequent damage on brain tissue (actually indicated by decreased metabolic activity in Fig. 1) and potentially also blood-brain-barrier and blood-CSF allow penetration of contrast agent and proteins into CSF irrespective of the primary cause. Therefore, the authors should provide more information when the hydrocephalus is first detectable and repeat the tracer experiments at these early time points.*

Can the authors measure the rate of CSF production in these mice?

The same problem with EM data in Figure 7. This is really intriguing and supports the authors' hypothesis.

However, such permeability changes need to be demonstrated at much earlier time points.

Responses: **(A)** The typical domed appearance of a hydrocephalus-harboring skull is apparent as early as P4, as shown in a new side-by-side comparison of pups at that age (Fig. 1A). **(B)** Though this is not stated in the MS, MR imaging of 3-week old pups is technically challenging. Since I prepared these pups for imaging together with my technician, I am intimately familiar with the experiment. It not only requires inserting a needle into the tail vein of a 3-week old pup (injecting the tail vein even of adult mice is considered a difficult task that requires training; see Groman and Reinhardt, J. Am. Assoc. Lab. Anim. Sci., vol. 43: pp. 35–38, 2004), but also keeping a catheter in place while the mouse is inserted into the bore of the MR machine, and throughout the imaging session that lasted more than 30 min, including the injection of the contrast medium which was done while the mouse was still in the machine. Furthermore, 3-week old hydrocephalus-harboring pups are about 35% smaller than their normal littermates (by weight, see Fig. 1B) to begin with.

Repeating this feat on a 2-week old *Mpdz* KO pup, for example, which weighs approximately 3.5 g, is practically impossible because the thinnest commercially-available syringe needle (gauge 34) is too thick to fit into its tail vein, and because its tail is too small for immobilizing a catheter. **(C)** To the best of my knowledge, the rate of CSF production in the mouse was reported for the first time only in June of this year (Steffensen et al., *Nature Comm.*, vol. 9: pp 1-13, 2018). That study measured the flow rate of a fluorescent probe administered directly into the lateral ventricles as an indicator of CSF production. The study established that the *Nkcc1* transporter located on the ventricle-facing surface of CP epithelial cells (CPECs) contributes significantly to CSF production. Therefore, we used its abundance as an indicator of CSF rate of production. Quantification of the new images we present in Fig. 10D shows that *Nkcc1* is approximately 75 percent more abundant in the CP of KO mice. **(D)** We analyzed by TEM the epithelial cell junctions in the CP of P14 pups, provided the images as expanded view data (Fig EV1), and discussed the subtler differences observed at that age between the junctions of WT and KO mice.

2. *Given the complexity in the in vivo setting the authors should try to demonstrate that deletion of Mpdz increases permeability of choroid plexus epithelial cells in vitro.*

Response: We provide now (Fig. 7E) impedance measurements of human papilloma CPECs (hpCPECs; the same cells used by Feldner et al., who referred to them as HIBCPP) in which *Mpdz* was depleted by transduction with shRNA. The results show that after reaching plateau, the impedance of the control group remained approximately 42 percent higher than that of the cells transduced by *Mpdz*-targeting shRNA. (Comment on the use of hpCPECs: though we ran experiments on the abundances of LDLR and of cell junction proteins on primary hCPECs, we switched to hpCPECs to reduce the cost of the functional experiment. Primary hCPECs and their specialty media cost approximately \$1000 per vial and can be passaged only up to 5 doublings; hpCPECs can be passaged more than 30 times. I believe that Feldner et al. used these cells for the same reason.)

3. *The H&E staining suggests an open aqueduct of Sylvius. However, more functional analysis of CSF flux through the ventricular system would be needed to exclude stenosis.*

Response: We present images (Fig. 5C) of Evans blue-injected WT and KO mice (the same technique used by Feldner et al.). They agree with the finding of Feldner et al. in regard to the presence of stenosis in the aqueduct. We revised the text accordingly. While, we don't rule out the possible contribution of aqueduct stenosis to the generation of hydrocephalus, this finding is not in conflict with the pathophysiological mechanism we propose.

4. *Figure 6: Why has only ZO-1 been analyzed? Mpdz binds to several TJ and AJ proteins and the literature suggests that loss of Mpdz does not lead to disassembly of junctions but rather to weakening. A detailed expression analysis of such proteins is needed otherwise such strong statements cannot be made.*

Response: We expanded the comparison of the abundance of CPEC junction proteins to a total of three (ZO1, JAMC, and E-CADHERIN). While we provide quantitative analysis of new confocal immunofluorescence (IF) images of these proteins in the CP (Fig. 6A-C), we opted also to use immunoblotting (IB) of MPDZ-depleted hCPECs for the same purpose, because is it more amenable to quantification (Fig. 6D), and because the amount of tissue in the CP of a 3-week old mouse is too small for IB. The IF and IB-based quantifications were in agreement with each other.

5. *Figure 8: Data on the LDLR are preliminary. A functional assay should be performed to show that loss of MPDZ affects LDLR-mediated endocytosis.*

Response: We compared LDLR endocytosis in MPDZ-depleted and in control hCPECs. Quantitative comparison of IF images (Fig. 9D) showed that LDLR was overabundant both on the surface and postendocytosis in MPDZ-deficient cells.

Minor comments:

Figure 3/4/5: Please, state the age of the animals and number of animals analyzed.

Figure 6/7: Number of biological replicates should be added.

Response: We added this information as suggested.

Referee 2

1. *Evidence to support altered transcytosis is not sufficient. EM images (TEM = transmission EM, not transmitted EM), may suffer from fixation problems, which commonly include formation of vacuoles. Quantification, sample numbers, ventricle location, and statistics are needed to make statements about junction length, for example. However as is, with the current approaches used and representative images shown, it is not possible to conclude that transcytosis is altered in this tissue. Classic approaches to examine transcytosis have not been used (e.g. Evans blue, HRP injections). Chow & Gu (Neuron 2017) provides one example on how some of these studies could be performed*

Response: I thank the reviewer for suggesting the protocol of Chow & Gu. We followed it and probed the CP of WT and KO mice by TEM. The new images (Fig. 8) show that the number of endocytose DAB particles was approximately 6-fold higher in the CPECs of *Mpdz* KO mice.

2. *Extensive evidence in the field demonstrates that the CSF is not a serum filtrate, and this is clear from any number of fine, contemporary reviews on the topic. In addition, given the novelty of proteomic CSF analysis in hydrocephalus, we have no reference point of comparison to understand the described characterizations and make a clear interpretation. How unique is the altered proteome to this model of hydrocephalus? Would hydrocephalus by any other mechanism also lead indirectly to heightened CSF ApoE and other protein expression (via elevated intracranial pressure or ventriculomegaly/distension, etc)? It seems the CSF is sampled after puncturing the lateral ventricle. This injury alone may induce changes in the CSF. Why not use a more conventional approach vetted by many groups for obtaining pure samples of CSF (e.g. DeMattos et al., J Neurochem 2002)?*

Response: **(A)** The Referee's point is well-taken. We removed the misleading first sentence of this section. Indeed, we attribute the overabundance of proteins in the CSF of the KO mice to increased transcytosis, not to an increase in paracellular permeability. **(B)** I thank the Referee for recognizing the novelty of our CSF proteomic analysis. Isn't absence of reference an inescapable consequence of novelty? However, we noted that out of the 313 proteins we identified, 295 had been detected in 3 previous studies on the CSF composition in the mouse. We interpreted the results by grouping functionally the 23 overabundant proteins in the CSF of KO mice. We discussed the significance of each group and suggested that "their overabundance is part of a multifaceted physiological response to the stress imposed on the brain by the expanding hydrocephalus". We hope that our data will serve as reference for future studies. I should point out, however, that at least one study (Finehout et al., Electrophoresis 2004, 25, 2564–2575) compared CSF of hydrocephalic patients and healthy subjects. However, Finehout et al. resolved the CSF by 2D gel chromatography and analyzed by mass-spectroscopy material extracted from gel spots, possibly reducing the detection sensitivity. Consequently, they detected only 82 proteins. We cite this reference in the revised MS. **(C)** I thank the Referee for supplying the above references. We extracted CSF through the cisterna magna of WT and KO mice and analyzed their proteomes. The differences between the CSF of WT and KO mice were of the same nature as between the previous CSF samples, i.e., overabundance of proteins in the CSF of KO mice, frequently of the same proteins as in the previous samples. We attribute the similarity of the compositions of the CSF samples extracted by the two methods to the careful technique we had used in the first round of experiments, and the minimal damage it incurred.

3. *Which choroid plexus is used in analyses? The lateral ventricles are the most severely affected, but most images shown seem to represent morphology of the 3rd ventricle choroid plexus. The age of sample collections may also influence interpretation of results. It appears that samples are typically collected and analyzed around P18, but by this age, the hydrocephalus is extensive and the authors state that the mice are not viable past ~P21 (3 weeks). It seems that hydrocephalus that is this extensive and induces severe injury in the brain and its vasculature, would alter the interstitial fluid and CSF. Hence, it is unclear if the effects described in the present study are primary, secondary, tertiary....including earlier pre-symptomatic or early hydrocephalic samples to the study would be informative.*

Response: Aside from the section shown in Fig. 6A, all the CP samples of KO mice were from the lateral ventricles. As seen in Fig. 5A and 5B, the 3rd ventricle CP is small and easy to miss. While

we interpret the composition of overabundant proteins in the CSF of the KO mouse as a stress response, we did not detect lesions, necrotic tissue, or bleeding in the brain parenchyma of KO mice at any age. We believe, therefore, that it is unlikely that brain injury contributed to the differences we identified in the CSF compositions of WT and KO mice. This argument was added to the Results section on CSF analysis.

4. What is the anatomical source of the contrast enhancement in Figure 4? The data suggests the choroid plexus might be a source, but the data presented shows a somewhat asymmetric contrast enhancement pattern (bottom of Figure 4B) and in other cases appears either highly focal and also in the aqueduct (top of Figure 4B). Given the presumed broad expression pattern of *Mpdz*, it is possible that alternative sources would include ependyma, circumventricular organs, or other potential areas of blood-brain barrier breakdown and vascular defects. Contrast enhancement might also suggest hemorrhage, which is not supported by the proteomic studies, but it would be worthwhile noting by presenting additional data regarding comparisons in content of basic serum proteins or albumin or by showing evidence for preserved vascularity of the choroid plexus. Lastly, what happens to contrast enhancement after 10min- does it dissipate in the CSF or persist in particular regions?

Response: **(A)** The anatomical source of the contrast medium had been highlighted in Fig. 4A and magnified in the insets in Fig. 4B. The clear fit between the locations of the CP in the T2-weighted images and the contrast medium brighter regions in the T1-weighted images leaves little doubt that the origin of the contrast medium is the CP. We revised the text of the legend to clarify this observation. **(B)** I thank the reviewer for suggesting a comparison of the serum albumin abundances in WT and KO CSF as a measure of BBB breakdown in the CP of the latter. There was no statistically significant difference between the WT and KO abundances of serum albumin as well as of three other serum proteins, ruling out breach of the BBB. This point was added to the text. We had presented TEM images of WT and KO CP capillaries to show they are morphologically similar. To strengthen this statement, we added magnified fields of the intercellular junctions and the fenestrae that are typical to these vessels. (Fig. EV2). **(C)** I am unable to respond to this question because we ended the MR imaging 10 min after the injection of contrast medium. The hydrocephalus-harboring pups were smaller by a 1/3 than their normal littermates. The T1-weighted contrast medium imaging followed 20 min of T2-weighted imaging of brain morphology. We were concerned that these pups will not survive a lengthy anesthesia and the other stresses involved in this experiment. Death of a mouse even at the very end of the 10 min T1-weighted contrast medium imaging would nullify the whole experiment and impede our overall progress because of the scarcity of KO mice.

5. Regarding the paracellular permeability mechanism and the presence of 23 new proteins in the CSF, what are the protein size and electrostatic charge comparisons? Do these proteins appear selected based on any criteria that could be more easily explained by ventricular permeability alone? What does the silver stain look like, and what is the typical CSF protein concentration in these samples?

Response: **(A)** We added the isoelectric points of all the proteins that were overabundant in the CSF of *Mpdz* KO mice to Table 1. **(B)** The possible connection between the condition of hydrocephalus and the identity of the overabundant CSF proteins had been discussed at length in the first version of the MS and in the current version. The sizes of these proteins range from 5 kDa (thymosin b-10) up to 509 kDa (apolipoprotein B-100) (Appendix Table A1). **(C)** We did not test the barrier function of the ependyma, but it is very unlikely to account for the protein overabundance in the CSF of *Mpdz*^{-/-} mice because Feldner et al., who used a very similar mouse model, reported that the ependymal tight junctions were morphologically normal. Though the pore size of tight junctions is not uniform, normally even the smallest CSF proteins are too large to pass through them (Shen et al., Annu. Rev. Physiol. 2011. 73:283–309). The most likely route of the CSF proteins is transcytosis, as discussed in the revised Results section. **(D)** The CSF was not resolved by SDS-PAGE. It underwent liquid chromatography followed directly by mass-spectroscopy, as described in the Methods section. **(E)** The mean protein concentrations in the CSF of WT and KO mice were 4.4 and 11.1 µg/mL, respectively, as shown in Fig. 10A of the revised MS.

6. It is unclear why the authors think the Feldner 2017 *Mpdz* knockout mouse has aqueductal stenosis, in contrast to the Milner 2015 mouse model? The discrepancy suggests that some of the

data may be unique to the Milner knockout model?

Response: Feldner et al. 2017 stated several times that the aqueduct of *Mpdz* KO mice was stenotic after P3. Their conclusion is based on tracking the flow of Evans blue dye injected into the right lateral ventricle, as shown in their Fig. 4A. Our new images of brains of WT and KO mice that underwent the same procedure agree with their findings (Fig. 5C). We revised the text accordingly. While, we don't rule out the possible contribution of aqueduct stenosis to the generation of hydrocephalus, this finding is not in conflict with the pathophysiological mechanism we propose.

Minor concerns

1. Why was PET imaging of the mice performed? There is no experimental justification for it as a starting experiment, and it does not offer any additional data that was not presented in the MRI data. This should be clarified, or the data moved to supplemental information.

Response: While PET does not achieve the sharp delineation between solid tissue and CSF that MRI provides, it distinguishes between glucose-consuming live tissue and non-cellular material. The observed match between the PET signal and the MR-resolved brain morphology shows that the MR-detected solid brain material was live brain parenchyma rather than necrotic tissue. We added text to this effect in order to explain the rationale of PET imaging. We believe that this information is significant because it rules out wide-spread tissue breakdown as an explanation of the large difference between the CSF compositions in WT and KO mice.

2. Figure 3- what was the n number of mice?

Response: Each row of one T2 and two T1-weighted MR images corresponds to one mouse. The same applies to Fig. 4. We clarified this in the legends of Figs. 3 and 4.

3. Figure 8 legend does not mention LDL-R

Response: We corrected this mistake.

4. Fig 3C mislabeled T2 and T1 in the legend. Fig4C legend also mislabeled T2 (should write T1)

Response: We corrected these mistakes.

5. "weighted" is misspelled on page 7 and in the legend of Fig 4

Response: We corrected these mistakes.

I thank the Referee for bringing to our attention these minor yet disconcerting mistakes.

2nd Editorial Decision

24 October 2018

Thank you for the submission of your revised manuscript to EMBO Molecular Medicine. We have now received the enclosed reports from the referees that were asked to re-assess it. As you will see the reviewers are now globally supportive and I am pleased to inform you that we will be able to accept your manuscript pending the following final amendments are addressed. Please provide a letter detailing your revision.

Please submit your revised manuscript within two weeks. I look forward to seeing a revised form of your manuscript as soon as possible.

I look forward to reading a new revised version of your manuscript as soon as possible.

2nd Revision - authors' response

6 November 2018

Referee 1:

The authors have addressed most of my comments in a satisfactory manner. I understand that the technical difficulties preclude some of the studies in younger animals.

Response: I appreciate the Referee's consideration of the technical limits I had described in my previous point-by-point reply.

Referee 2:

The authors have made remarkable revisions to the original manuscript and taken steps to address concerns raised, which substantially strengthen their study.

It would be preferable for analyses of the transcytosis and immunostaining to be performed at much earlier ages, as suggested, so at late embryonic or early postnatal ages when the hydrocephalus first presents in mice. That said, the work, time, and breeding schemes required for replicating these studies in younger animals at this point may not be feasible. Hopefully, the authors will consider this suggestion in future studies/analyses of these mice and will include this point in the discussion.

Response: I appreciate the Referee's recognition of the work my lab and collaborators invested in the study. The Referee's point in regard to earlier time points is well-noted and will be one of our short-term objectives.

This reviewer is not convinced that the authors have thoroughly examined the anatomical locations of choroid plexuses used for their experiments. In the revision, they state that the majority of images are from the lateral ventricle choroid plexus. This does not seem consistent with the images as presented. For example, the current Figure 4 choroid plexus looks to be from the 3rd ventricle.

Response: The Referee is correct in identifying the images of the CP shown in Fig. 4A as coming from the 3rd ventricle. This is exactly what I stated in my previous point-by-point reply. All the other images of *Mpdz*^{+/+} and *Mpdz*^{-/-} CPs were from the lateral ventricles. To clarify this point, the location of all the CP samples shown in MS are identified in the revised text and figure legends.

Corresponding Author Name: Dr. Arie Horowitz

Manuscript Number: EMM-2018-09540